# Replicable Online pricing

**Kiarash Banihashem**
University of Maryland
College Park, MD, USA
kiarash@umd.edu

**MohammadHossein Bateni**
Google Research
New York City, New York, USA
bateni@google.com

**Hossein Esfandiari**
Google Research
London, UK
esfandiari@google.com

**Samira Goudarzi**
University of Maryland
College Park, MD, USA
samirag@umd.edu

**MohammadTaghi Hajiaghayi**
University of Maryland
College Park, MD, USA
hajiaghayi@gmail.com

## Abstract

We explore the concept of replicability, which ensures algorithmic consistency despite input data variations, for online pricing problems, specifically prophet inequalities and delegation. Given the crucial role of replicability in enhancing transparency in economic decision-making, we present a replicable and nearly optimal pricing strategy for prophet inequalities, achieving a sample complexity of $\text{poly}(\log^* |\mathcal{X}|)$, where $\mathcal{X}$ is the ground set of distributions. Furthermore, we extend these findings to the delegation problem and establish lower bound that proves the necessity of the $\log^* |\mathcal{X}|$ dependence. En route to obtaining these results, we develop a number of technical contributions which are of independent interest. Most notably, we propose a new algorithm for a variant of the heavy hitter problem, which has a nearly linear dependence on the inverse of the heavy hitter parameter, significantly improving upon existing results which have a cubic dependence.

## 1 Introduction

Scientific research fundamentally relies on the ability to repeat experiments and achieve similar results. This principle, known as "reproducibility" is critical to validating findings and ensuring progress across fields. However, many researchers [HIB+18, IHGP17, LKM+18] have raised concerns about a "reproducibility crisis" where studies often fail to replicate reliably. This has highlighted the pressing need for methods that guarantee reproducibility and transparency in research.

A significant challenge in achieving reproducibility is the inherent variability in data, which often stems from complex and stochastic processes. Even when researchers meticulously document their methods, others may struggle to reproduce the results due to this randomness. This variability opens the door to misleading findings, whether intentional or unintentional. For instance, practices such as "$p$-hacking"—where researchers test multiple hypotheses until a significant result is found—can undermine the validity of scientific conclusions.

To address these challenges from a theoretical perspective, Impagliazzo et al. [ILPS22] (STOC'22) introduced the notion of *replicable algorithms* for statistical problems.[1] Formally, an algorithm $\mathcal{A}$ that takes as input a set of samples $S \subseteq \mathcal{X}^n$ and a binary string $r$ representing its internal randomness is called $\rho$-*replicable* if, for any distribution $\mathcal{D}$ over $\mathcal{X}$,

$$\Pr_{S_1, S_2 \sim \mathcal{D}, r \sim R} [\mathcal{A}(S_1; r) = \mathcal{A}(S_2; r)] \geq 1 - \rho, \tag{1}$$

---

[1]The original work used the term "reproducible" (to align with common usage), but subsequent works have adopted "replicable", the term we use here.

39th Conference on Neural Information Processing Systems (NeurIPS 2025).

where $R$ denotes the distribution of the random bits used by the algorithm. Intuitively, Equation (1) ensures that the algorithm's output is determined primarily by the input distribution rather than specific samples or randomness. This safeguards against data manipulation or corruption, as such actions are easily detectable by re-running the algorithm. Bun et al. [BGH+23] (STOC'23) further developed the theory of replicable algorithms by relating it to other notions of stability such as differential privacy, adaptive generalization. Replicable algorithms have been developed for numerous problems including high dimensional mean estimation [HIK+24b] (FOCS'24), clustering [EKM+23] (NeurIPS'23), bandits [EKK+23] (ICLR'23), and learning halfspaces [KKL+24] (ICML'24).

Replicability is especially critical in settings where decisions have significant real-world consequences, such as public systems, government policies, or large-scale markets. In these scenarios, fairness and transparency are paramount. Even when the decision-making process is publicly accessible, decision-makers could manipulate outcomes by selectively choosing data to support desired conclusions. Replicability acts as a certificate that the decision-making process is robust and immune to such manipulations, particularly in cases where the decision space is large or continuous, and no single "best" choice exists.

Motivated by these considerations, we study replicability in the context of the *prophet inequalities* problem, a fundamental model in online decision-making with well-established significance from both theoretical and economic perspectives [HKP04, BIK07, HKS07, CHMS10]. In its simplest form, this problem involves a sequence of random values $X_1, \ldots, X_N$ arriving online, where each $X_i$ is drawn from a known distribution $D_i$. The goal is to select a single value that is large in expectation. A classic solution involves using a fixed threshold $\tau$ and accepting the first value that exceeds $\tau$. It is well-known (e.g., see [KW12]) that setting $\tau = \mathbb{E}\left[\max\{X_1, \ldots, X_N\}\right]/2$ guarantees, in expectation, a value at least half of the offline optimum $\mathbb{E}\left[\max\{X_1, \ldots, X_N\}\right]$, which is tight. Similarly, setting $\tau$ to be the median of $\mathbb{E}\left[\max\{X_1, \ldots, X_N\}\right]$, with appropriate tie-breaking when the distribution has atoms, leads to the same $1/2$ guarantee [SC84].

While the classical formulation assumes that the distributions $D_1, \ldots, D_n$ are fully known, this is often unrealistic in practice. Instead, we consider a sample-based framework, where we only have access to a finite number of samples from each $D_i$. The key challenge here is to design a replicable algorithm that uses few samples to set a threshold $\tau$ achieving a competitive performance. We say a threshold is $\alpha$-competitive if the corresponding strategy obtains at least $\alpha \mathbb{E}\left[\max\{X_1, \ldots, X_N\}\right]$ in expectation. An algorithm solves the *Replicable Online Pricing (ROP)* problem with parameters $(\alpha, \rho, \beta)$ if it is $\rho$-replicable and outputs an $\alpha$-competitive threshold with probability at least $1 - \beta$. In order to avoid issues with tie-breaking, we use an extra "coin" to decide whether an element with value equal to $\tau$ should be accepted (see Section 2 for further details on this formulation). In this work, we analyze the sample complexity of the ROP problem, presenting efficient algorithms and proving impossibility results.

We further study the implication of our results to the *delegation problem*, which has been widely studied in economics under various settings and models [Hol80, FJ83, CF98, AV10, AB13, AE17, BGH01, KK18, HMRS24, BHHS24]. At its core, delegation involves an authority figure, referred to as *the principal*, facing a challenge and relying on an expert, referred to as *the agent*, to solve it, with the agent being responsible for searching for and proposing a solution based on their expertise. A practical example of this is the relationship between a public entity and a private contractor or the intra-organizational workflow between corporate management and a specialized division. Each possible solution yields specific utilities for the principal and the agent, often leading to utility misalignment due to differing incentives. To mitigate this misalignment and ensure the consideration of their interests, the principal may announce a set of acceptable solutions (e.g., by establishing certain criteria) before the agent begins their search. This preemptive strategy helps enforce alignment between the principal's objectives and the agent's proposals, balancing the trade-off between granting the agent enough autonomy and maintaining control over outcomes that align with the principal's goals.

In particular, we consider a model similar to the model studied by [KK18] (EC'18). We assume there is a distribution $\mathcal{D}$ defined over an abstract space $\Omega$ of possible solutions for the delegated task. Each solution $\omega \in \Omega$ has an associated quality from the principal's perspective, denoted by $x(\omega)$, and a possibly different quality from the agent's perspective, denoted by $y(\omega)$. The agent is supposed to make $N$ independent and identically distributed (i.i.d.) draws from the distribution $\mathcal{D}$, resulting in a collection of $N$ candidate solutions $\omega_1, \omega_2, \ldots, \omega_N \in \Omega$. The agent then examines $x(\omega_i)$ and $y(\omega_i)$

for each candidate $\omega_i$ and then chooses one to present to the principal. As mentioned, in the absence of any constraints from the principal, the agent would trivially select the solution $\omega_i$ maximizing their own objective $y(\omega_i)$, resulting in the principal receiving the corresponding value of $x(\omega_i)$. To enhance this arrangement, the principal imposes a constraint at the outset, specifying that they will only accept solutions $\omega$ for which $x(\omega)$ exceeds a certain threshold $\tau$. As in the ROP problem, we allow the use an extra coin for tie-breaking. (see Section 2 for more details) This causes the agent to pick the $\omega_i$ maximizing $y(.)$ subject to the acceptance constraint.

We assume that the principal can learn about $\mathcal{D}$ by drawing samples and observing their associated quality value $x$ and can use its knowledge in devising its price. The principal's goal is to set a price such that the obtained solution provides a good approximation to the optimal $\omega_i$ for the principal. In particular, we say a threshold $\tau$ is $\alpha$-competitive if the expected utility of the principal using $\tau$ is at least $\alpha\mathbb{E}\left[\max\left\{x(\omega_1),\ldots,x(\omega_N)\right\}\right]$. An algorithm solves the *Replicable Delegation (RD)* problem if it is $\rho$-replicable and with probability at least $1-\beta$ outputs an $\alpha$-competitive threshold.

## 1.1  Our contribution

Our first result is an algorithm that replicably outputs a $(1/2-\epsilon)$-competitive threshold with high probability for finite distributions. A standard approach to prophet inequalities is to set $\tau = \mathbb{E}\left[\max\left\{X_1,\ldots,X_N\right\}\right]/2$, suggesting that one might attempt to apply this strategy to the ROP problem using a replicable mean estimation algorithm, such as that of [ILPS22], to approximate this threshold. However, as we discuss in Section 3, these algorithms have an additive error, preventing them from yielding competitive prices.

To overcome this limitation, we design an algorithm based on the median approach of [SC84], leading to the following result.

**Theorem 1.** *Assume that the distributions $D_i$ are all supported on a known finite set $\mathcal{X}$. For any $\epsilon \in (0,1/2)$ and $\rho,\beta > 0$, there exists an algorithm solving the ROP problem with parameters $(1/2-\epsilon,\rho,\beta)$ using at most* $\mathrm{poly}\left(\log^* |\mathcal{X}|, \rho^{-1}, \beta^{-1}, \epsilon^{-1}\right)$ *samples from each $D_i$.*

To achieve the above result, we show that good prices can be obtained by applying replicable (approximate) median estimation algorithms to the distribution of $\max\left\{X_1,\ldots,X_N\right\}$. Such algorithms have previously been obtained by [ILPS22] and [BGH+23] [2]. While both of these algorithms have an approximation error in estimating the median, unlike the algorithms for mean estimation, the effect of these errors on the quality of the output price can be bounded. As we show in Lemma 4, a $\gamma$-approximate median (see Section 2 for a formal definition) leads to a $(1/2-\gamma)$-competitive price. We further generalize our technique and apply it to the delegation problem, obtaining the following result.

**Theorem 2.** *Assume that the distribution $\mathcal{D}$ is supported over a known finite set $\mathcal{X}$. For any $\epsilon \in (0,1/2)$ and $\rho,\beta > 0$, there exists an algorithm solving the RD problem with parameters $(1/2-\epsilon,\rho,\beta)$ using at most $N \cdot \mathrm{poly}\left(\log^* |\mathcal{X}|, \rho^{-1}, \beta^{-1}, \epsilon^{-1}\right)$ samples from $\mathcal{D}$.*

For the ROP problem, we establish that the dependence on the size of $\mathcal{X}$ is necessary. Formally, letting the sample complexity of an algorithm denote the number of samples it requires from each $D_i$ (see Definition 1), we prove the following theorem.

**Theorem 3.** *There exist constants $\rho,\beta,c_1,c_2 > 0$ and a set $\mathcal{X}$ with the following property. Fix $\alpha > c_1(\log^* |\mathcal{X}|)^{-0.99}$ and $N \geq \frac{c_2}{\alpha}$. Any algorithm that can solve the $(\alpha,\rho,\beta)$-replicable online pricing problem has sample complexity $\widetilde{\Omega}(\alpha \log^* |\mathcal{X}|)$, even if we assume that $X_1,\ldots,X_N$ are i.i.d.*

The constant 0.99 in the theorem above can be replaced with any value strictly less than 1. To prove this theorem, we reduce from the replicable *interior point problem*, where the goal, given a distribution $D$, is to output a value in the range $[\min(D),\max(D)]$ in a replicable manner (see Section 2). By leveraging the privacy-to-replicability reduction of [BGH+23], we establish that this problem requires at least $\widetilde{\Omega}(\log^* |\mathcal{X}|)$ samples (see the Appendix).

---

[2]Specifically, we use the algorithm of [12], which has a polynomial sample complexity. However, we note that their algorithm is not computationally efficient due to reliance on correlated sampling. Alternatively, one can use the approach of [31] which is polynomial time but has a sample complexity exponential in $\log^* |\mathcal{X}|$. We note that since our result is obtained via a reduction to replicable median estimation (see Lemma 8), any possible future improvements for that problem would immediately yield improvements to our results as well.

Moreover, we show that the "hard distributions" for the interior point problem exhibit an additional structural property: they contain no "heavy" points. Specifically, for any $\alpha \geq \Omega(\log^* |\mathcal{X}|)^{-0.99}$, no point in the distribution has probability mass exceeding $\alpha$. We then argue that, in such instances, any competitive price must lie within the interior of the distribution of $X_i$. This is proved in Lemma 8 and the main intuition behind it is as follows. If the price is set outside this interior, the algorithm will either reject all elements or accept only the first element. Using the fact that the distribution contains no points with probability exceeding $\alpha$, we can construct instances of the prophet inequalities problem where, for an appropriate range of $N \approx \alpha^{-1}$, the first element is roughly $1/N$ times the maximum element. Therefore, if the price is outside the interior of the distribution, its competitive ratio is at most $\approx 1/N$. We then extend this result to arbitrary $N$ by modifying these instances to output 0 with a small probability. Further details can be found in Section 4.

We further show that, in the special case of i.i.d. distributions, we can match the dependence on $\alpha$ in theorem below.

**Theorem 4.** *For any $\rho, \beta > 0$ and $\alpha \leq 1/4$[3], there is an algorithm that can solve the ROP pricing problem for i.i.d. distributions with parameters $(\alpha, \rho, \beta)$ and sample complexity*

$$O(\alpha \cdot \mathrm{poly}(\log^* |\mathcal{X}|, \rho^{-1}, \beta^{-1}))$$

En route to proving our results, we develop a novel and improved replicable algorithm for outputting a heavy hitter for an input distribution. A value $x$ is called a heavy $\nu$-heavy hitter for a distribution $D$ if its sampling probability under $D$ is at least $\nu$. In the heavy hitter problem, we are given a distribution $D$ over a finite set $\mathcal{X}$, which is not necessarily ordered, and the goal is to (replicably) output a $\nu$-heavy hitter for $D$. We focus on an approximate version of the problem which assumes that the underlying distribution is guaranteed to have a $(\gamma\nu)$-heavy hitter. For this problem, we obtain the following result.

**Theorem 5.** *For any $\rho, \beta, \nu > 0$ and $\gamma \geq 4$, there exists a $\rho$-replicable algorithm with sample complexity $O\left(\nu^{-1}\rho^{-2}\log^3(\nu^{-1})(\log(\rho^{-1}) + \log(\beta^{-1}))\right)$ with the following guarantee: Assuming a $(\gamma\nu)$-heavy hitter exists, the algorithm outputs a $\nu$-heavy hitter with probability at least $1 - \beta$. If no $(\gamma\nu)$-heavy hitter exists, then with probability at least $1 - \beta$, it outputs either* `Null` *or a $\nu$-heavy hitter.*

Existing work providing replicable algorithms for the heavy hitter problem consider a variant where the goal is to return a list that contains all $(\gamma\nu)$-heavy hitters and is a subset of $\nu$-heavy hitters. For this variant, Impagliazzo et al. [ILPS22] obtain a $\rho$-replicable algorithm with $1 - \beta$ success probability that has sample complexity $\widetilde{O}(\frac{1}{\min(\rho,\beta)^2\nu^4(\gamma-1)^2})$. Esfandiari et al. [EKM+23] improve on this with an algorithm achieving sample complexity $\widetilde{O}(\frac{1}{\rho^2\nu^3(\gamma-1)^2}\log(1/\beta))$. More recently, Hopkins et al. [HIK+24b] developed an algorithm with expected sample complexity $O(\frac{1}{\rho\nu^3(\gamma-1)^2}\log(\frac{1}{\min(\beta,\rho)\nu}))$. All of these results have a cubic dependence on $\nu$, whereas our approach achieves a nearly linear dependence. Although, it is worth noting that the variant considered by the previous works is more general than the one we consider here. We note that, even without replicability, one requires at least $\Omega(\nu^{-1})$ samples to find a heavy-hitter and as such our result is nearly tight. We further note that without the improved dependence, the range of $\alpha$ in Theorem 3 would be $\Omega((\log^* |\mathcal{X}|)^{-0.33})$ instead of $\Omega((\log^* |\mathcal{X}|)^{-0.99})$.

To prove the above result, we design a two-stage algorithm. First, we sample a candidate set $S^{(1)}$ of size approximately $\nu^{-1}$. Then, using a fresh set of samples $S^{(2)}$, we estimate the probability of each point in $S^{(1)}$. The elements in $S^{(1)}$ are randomly permuted, and we output the first element (in the order of the permutation) whose estimated probability exceeds a threshold $\nu'$, drawn uniformly from $[\nu, \gamma\nu]$.

A key challenge in implementing this strategy is ensuring that the random permutation is defined replicably. This is nontrivial, since the permutation depends on $S^{(1)}$, which itself depends on the samples. To address this, we use $k$-wise independent sampling based on Reed-Solomon codes to assign a number to each element in $\mathcal{X}$, then sort $S^{(1)}$ based on these numbers. Crucially, while the numbers are not fully independent, any subset of size $|S^{(1)}|$ behaves as if chosen independently.

---

[3]Note that one can handle $\alpha \in (1/4, 1/2)$ using Theorem 1

Although the two-stage approach has been previously explored (e.g., see [EKM$^+$23]), our tailored analysis significantly reduces the sample complexity from cubic in $\nu^{-1}$ to nearly linear. This improvement stems from two key insights. First, our use of random permutation reduces the number of "important" candidates (from a replicability perspective) from $\nu^{-1}$ to $\log(\nu^{-1})$. Second, we leverage a sharper version of the Chernoff inequality, yielding an additional factor of $\nu^{-1}$ in savings. Importantly, this sharper version can only be leveraged for our problem, and not the more general variant studied by Esfandiari et al. [EKM$^+$23].

**Map of the paper.** The remainder of the paper is organized as follows. Section 2 discusses the preliminaries of the paper. Section 3 proves the upper bound for ROP. Section 4 proves the lower bound. Finally, Section 5 provides the heavy-hitter algorithm. Due to space constraints, parts of the proofs, the proof for RD, as well as the discussion of further related work is deferred to the Appendix.

## 2 Preliminaries

**Notation and setting.** Given a positive integer $n$, we use $[n]$ to denote the set $\{1, \ldots, n\}$. The prophet inequality problem is defined as follows. $N$ random values $X_1, \ldots, X_N$ arrive in an online manner, where $X_i$ is sampled from a distribution $D_i$ and the goal is to choose a single item with high value. The distributions $D_i$ are all supported over a set $\mathcal{X} \subseteq \mathbb{R}^{\geq 0}$ known upfront.

We will focus on *threshold based algorithms* for this problem; given a fixed price $\tau \in \mathbb{R}^{\geq 0}$ we define $i(X_1, \ldots, X_N; \tau, 0) = \arg\min\{i : X_i \geq \tau\}$ and $i(X_1, \ldots, X_N; \tau, 1) = \arg\min\{i : X_i > \tau\}$ to denote, respectively, the index of the first value greater than or equal to $\tau$ and the index of the first value strictly greater than $\tau$. If no such value exists, we set $i(.)$ to $+\infty$. For simplicity, we will often write $i(\tau, 0)$ and $i(\tau, 1)$. Let $\kappa \in [0, 1]$ and sample $B \sim \text{Bernoulli}(\kappa)$. We say a *pricing pair* $(\tau, \kappa)$ is $\alpha$-competitive for $D_1, \ldots, D_N$ if $\mathbb{E}\left[X_{i(\tau, B)}\right] \geq \alpha \mathbb{E}\left[\max\{X_1, \ldots, X_N\}\right]$, where the expectation is over the randomness of $X_1, \ldots, X_N, B$. We rely on tie-breaking because our median-based pricing requires the algorithm's acceptance probability to be close to $1/2$. If $\max X_1, \ldots, X_N$ has a large atom at its median, no single threshold $\tau$ achieves this without tie-breaking. Although one of $\mathbb{E}\left[X_{i(\tau, 0)}\right]$ or $\mathbb{E}\left[X_{i(\tau, 1)}\right]$ must match the $\alpha$-competitiveness of $\mathbb{E}\left[X_{i(\tau, B)}\right]$, identifying which one requires full knowledge of the distributions. Intuitively, if $X_i$ occasionally takes very large values, $B = 1$ is preferable; otherwise, $B = 0$ is better, but such rare events cannot be detected from samples.

One can define a replicable variant of the prophet inequalities problem by requiring that the algorithm outputs a price in a $\rho$-replicable manner while maintaining an $\alpha$-competitive guarantee. That is, in expectation over both the randomness of $X_i$ and the algorithm itself, the expected output must be at least $\alpha \mathbb{E}\left[\max\{X_1, \ldots, X_N\}\right]$. However, this formulation has a fundamental issue: an algorithm can effectively bypass the replicability requirement by outputting 0 with probability $1 - \rho$ and, with probability $\rho$, employing a non-replicable $1/2$-competitive algorithm (e.g., the single-sample algorithm from Rubinstein, Wang, and Weinberg [RWW19]). This approach achieves a $\rho/2$-competitive ratio, which is constant for constant $\rho$, while still technically satisfying replicability. Intuitively, though, such an approach is undesirable because its competitiveness derives entirely from the non-replicable outputs.

Given the above issue, we opt for the following formulation, which requires the output price itself to be $\alpha$-competitive.

**Definition 1** (Replicable Online Pricing (ROP)). *An algorithm $\mathcal{A}$ solves the Replicable Online Pricing (ROP) problem with parameters $(\alpha, \rho, \beta)$ and sample complexity $n$ if it takes $n$ samples from each distribution $D_i$, is $\rho$-replicable, and outputs an $\alpha$-competitive price pair $(\tau, \kappa)$ with probability at least $1 - \beta$.*

This definition aligns with existing work on replicability in statistical problems, wherein the quality of an algorithm's output is assessed by defining an "acceptable set" (in this case, the set of all $\alpha$-competitive prices) and requiring that the algorithm's output falls within this set with high probability.

The *Replicable Delegation(RD)* problem is defined analogously. In the delegation problem, an agent samples solutions $\omega_1, \ldots, \omega_N$ from a distribution $\mathcal{D}$, where each sample has utility $x(\omega_i)$ and $y(\omega_i)$ for the principal and the agent respectively. The principal commits to a price $\tau$, which is announced to the agent beforehand, and only accepts solutions satisfying $x(\omega_i) \geq \tau$ or $x(\omega_i) > \tau$. The agent then chooses the acceptable solution maximizing its own utility. Let $i(\tau, B) = \arg\max_{i : x(\omega_i) \geq \tau} y(\omega_i)$

for $B = 0$ and $i(\tau, B) = \arg\max_{i:x(\omega_i) > \tau} y(\omega_i)$ for $B = 1$. We say the pricing pair $(\tau, \kappa)$ is $\alpha$-competitive if $\mathbb{E}\left[\omega(x_{i(\tau, B)})\right] \geq \alpha\mathbb{E}\left[\max\{x(\omega_1), \ldots, x(\omega_N)\}\right]$, where $B \sim \text{Bernoulli}(\kappa)$. An algorithm $\mathcal{A}$ solves the RD problem with sample complexity $n$ if it is $\rho$-replicable, takes $nN$ samples from $D$ and with probability $1 - \beta$ outputs an $\alpha$-competitive price. Note that we allow $nN$ samples here, instead of just $n$, in order to align with the ROP definition.

Throughout the paper, we will frequently use the following multiplicative form of the standard Chernoff bound. Let $p \in [0, 1]$ and sample $n$ i.i.d variables $Y_1, \ldots, Y_n$ from the distribution $\text{Bernoulli}(p)$. define $\hat{p} = \frac{\sum_i Y_i}{n}$.

$$\Pr\left[|\hat{p} - p| \geq \delta p\right] \leq 2e^{-\delta^2 np/3} \tag{2}$$

**Interior point and approximate median.** Given a distribution $D$ over an ordered set $\mathcal{X}$, we say $x$ is in the interior point of $D$ if $\min(D) \leq x \leq \max(D)$ where $\min(D)$ and $\max(D)$ denote, respectively, the minimum and maximum elements with strictly positive probability under $D$. We say $x$ is a $\gamma$-approximate median for $\gamma < 1/2$ if $\min\{\Pr_{X \sim D}[X \geq x], \Pr_{X \sim D}[X \leq x]\} \geq 1/2 - \gamma$. The two problems are closely connected; any approximate median is automatically an interior point and, conversely, an algorithm for (replicable) interior point can be transformed into an algorithm for (replicable) approximate median with a slight increase in sample complexity. As shown by [BGH$^+$23], there exists an algorithm for the replicable interior point problem with sample complexity polynomial in $\log^* |\mathcal{X}|$.

**Lemma 2.** *For any $\rho, \beta, \gamma > 0$, there exists a $\rho$-replicable algorithm that outputs a $\gamma$-approximate median with probability at least $1 - \beta$ and has sample complexity $poly(\log^* |\mathcal{X}|, \rho^{-1}, \beta^{-1}, \gamma^{-1})$.*

We refer to [BGH$^+$23] for the proof. Specifically, following the argument in Footnote 18 of the arXiv version of [BGH$^+$23] and keeping track of the relevant parameters, we obtain the sample complexity $\widetilde{O}\left(\varepsilon^{-4}\rho^{-2}\log^2(1/\beta)(\log^* |\mathcal{X}|)^3\right)$.

It is clear that if a distribution has a large atom (e.g., if it takes a single value with probability 1), then $\Pr_{X \sim D}[X \geq x]$ can never be close to $1/2$, which is required for median pricing. To address this, we extend the definition of approximate median to pricing pairs $(\tau, \kappa)$. We say a pricing pair $(\tau, \kappa)$ is a $\gamma$-approximate median for $D$ if

$$(1 - \kappa)\Pr_{X \sim D}[X \geq \tau] + \kappa\Pr_{X \sim D}[X > \tau] \in (1/2 - \gamma, 1/2 + \gamma).$$

We further prove the following result in the Appendix.

**Proposition 3.** *For any $\rho, \beta, \gamma > 0$, there is a $\rho$-replicable algorithm that outputs a $\gamma$-approximate median pair $(\tau, \kappa)$ with probability at least $1 - \beta$ and has sample complexity $poly(\log^* |\mathcal{X}|, \rho^{-1}, \beta^{-1}, \gamma^{-1})$.*

## 3 Replicable Online Pricing Upper Bound

In this section, we prove Theorem 1 and Theorem 4 via a reduction to the replicable (approximate) median estimation problem. As mentioned earlier, one natural approach to solving the ROP problem is to use existing replicable mean estimation algorithms to compute $\mathbb{E}[\max\{X_1, \ldots, X_N\}]/2$. This may initially seem more appealing than our median-based approach, as mean estimation algorithms do not require a finite ground set $\mathcal{X}$ and even work for continuous distributions, whereas median estimation incurs a dependence on $\log^* |\mathcal{X}|$. However, these approaches introduce additive estimation error, which can be problematic when $\mathbb{E}[\max\{X_1, \ldots, X_N\}]$ is small, potentially leading to negative outputs. While one might attempt to mitigate this issue using techniques such as adjusting the error based on an initial sample set, our lower bound in Theorem 3 establishes that no approach—including these—can eliminate the dependence on $\log^* |\mathcal{X}|$.

We first state the following lemma which follows folklore techniques (e.g., see [KK18]). In the interest of space, we have moved the proof to the Appendix.

**Lemma 4.** *Let $(\tau, \kappa)$ be a $\gamma$-approximate median pair for the distribution of $\{X_1, \ldots, X_N\}$. Then $(\tau, \kappa)$ is a $(1/2 - \gamma)$-competitive price pair.*

*Proof of Theorem 1.* Let $s = \text{poly}(\log^* |\mathcal{X}|, \rho^{-1}, \beta^{-1}, \epsilon^{-1})$ denote the sample complexity specified by Proposition 3. For each $1 \leq i \leq n$, we take $s$ samples from the random variable $X_i$ with

distribution $\mathcal{D}_i$, denoting these samples by $x_{i,1}, \ldots, x_{i,s}$. Then, for each $1 \leq j \leq s$, we define $y_j := \max_{1 \leq i \leq n} x_{i,j}$. It follows that $y_1, \ldots, y_s$ are $s$ samples taken independently from the random variable $Y$ defined as $Y := \max(X_1, \ldots, X_N)$.

Using the given $\rho$-replicable algorithm for $\gamma$-approximate median pair $(\tau, \kappa)$, we compute a $\gamma$-approximate median of $Y$ based on the samples $y_1, \ldots, y_s$. We return this as the output of our pricing algorithm, which we know is $(1/2 - \gamma)$ competitive by Lemma 4, and therefore satisfies the desired properties of the $(1/2 - \gamma, \rho, \beta)$ online replicable pricing algorithm. Hence, the theorem follows. □

*Proof of Theorem 4.* We first observe that since the elements are i.i.d,

$$\mathbb{E}[X_1] = \frac{1}{N} \mathbb{E}[X_1 + \ldots, X_N] \geq \frac{1}{N} \mathbb{E}[\max\{X_1, \ldots, X_N\}].$$

We therefore assume that $\alpha \geq 1/N$ as otherwise, using the price $\tau = 0$, we obtain a $(1/N)$-competitive price.

Set $r = \lfloor \frac{1}{4\alpha} \rfloor$. Since $\alpha \geq 1/N$, we have $r \leq N$. Divide the input variables $X_1, \ldots, X_N$ into groups of size $\lceil N/r \rceil$, with the last group possibly having less than $\lceil N/r \rceil$ elements. Let $(\tau, \kappa)$ be a 1/4-approximate median pair for $\max\{X_1, \ldots, X_{\lceil N/r \rceil}\}$. We estimate this replicably as in Proposition 3. By Lemma 4, using this pair the expectation of the output is at least $\frac{1}{4} \mathbb{E}[\max\{X_1, \ldots, X_{\lceil N/r \rceil}\}]$. Since we have at most $r$ groups however,

$$\mathbb{E}[\max\{X_1, \ldots, X_N\}] \leq r \mathbb{E}[\max\{X_1, \ldots, X_{\lceil N/r \rceil}\}].$$

Therefore, $(\tau, \kappa)$ is $\frac{1}{4r}$-competitive. Since $r \leq \frac{1}{4\alpha}$, this means the price it at least $\alpha$-competitive.

We next bound the sample complexity of our algorithm. To calculate $(\tau, \kappa)$, it suffices to have $\text{poly}(\log^* |\mathcal{X}|, \rho, \beta)$ samples from $\max\{X_1, \ldots, X_{\lceil N/r \rceil}\}$. Observe however that we always have at least $r/2$ full groups; if $r = 1$ then we have exactly $r$ full groups and if $r \geq 2$ we have at least $r - 1 \geq r/2$ full groups. Therefore, it suffices to have $\frac{2}{r} \text{poly}(\log^* |\mathcal{X}|, \rho^{-1}, \beta^{-1})$ samples from each of the input variables. Since $r = \lfloor \frac{1}{4\alpha} \rfloor$ and $\alpha \leq 1/4$, we have $r \geq \frac{1}{8\alpha}$, which means $\frac{2}{r} \leq 16\alpha$. Therefore, the sample complexity bound follows as well. □

## 4 Replicable Online Pricing Lower bound

In this section we prove Theorem 3. Our proof is obtained via a reduction from the interior point problem. Specifically, we rely on the following result.

**Proposition 5.** *There exists $\rho, \beta > 0$ such that any $\rho$-replicable algorithm solving the statistical interior point problem with failure probability at most $\beta$ for a distribution $Z$ over a finite set $F$ has sample complexity $\widetilde{\Omega}(\log^* |F|)^4$.*

The above result follows from the hardness of the differentially private version of the problem; we refer to Appendix E.2 for more details.

We note that the nature of the set $F$ do not play a major role in the problem beyond its size; for any other ordered set $F'$ of the same size, an algorithm for the set $F$ can be turned into an algorithm for the set $F'$ and vice versa. Our proof of Theorem 3 is divided into multiple steps. Due to space constraints, we sketch the steps and state the Lemmas here and refer to the Appendix for the full proofs.

1. We prove that the lower bound for the interior point problem holds even under the additional assumption that the underlying distribution contains no "heavy" elements—those with sampling probability exceeding $\alpha$, where $\alpha$ falls within the range specified in Theorem 3 (Lemma 7). To prove this, we show that if the distribution did contain heavy elements, the statistical interior point problem could be solved using a novel heavy hitter algorithm, which we introduce in Section 5. This effectively implies that the "hard instances" for the replicable statistical interior point problem exclude heavy elements.

---

[4]This lemma also implies that this problem is unsolvable with any finite sample complexity over infinite domains since a solution for an infinite set would yield a solution for arbitrarily large finite subsets, which by this lemma requires arbitrarily large sample complexity.

2. We next show that, for a suitable choice of the set $\mathcal{X}$, if the distribution $X$ does not contain a heavy element with probability $\geq \alpha$, then the maximum of $M \approx 1/\alpha$ i.i.d. draws of $X$ is, in expectation, at least $\Omega(M)\mathbb{E}[X]$ (Lemma 8). We refer to distributions without such heavy elements as $\alpha$-*light*.

3. We then prove that any $\alpha$-light instance for the interior point problem can be transformed into an instance of online pricing with $N \geq \Omega(1/\alpha)$ such that improving on $\Theta(\alpha)$-competitive prices can only be done by solving the interior point problem. In this reduction each sample from an $X_i$ in online pricing corresponds to roughly $1/\alpha$ samples for interior point (Lemma 9).

We proceed with a formal proof. We first extend the above hardness result to the case where the distribution does not contain any heavy elements. We call such distributions *light* and formally define them below.

**Definition 6** (light distribution). *A distribution $Z$ is $\alpha$-light if $\Pr[Z = z] \leq \alpha$ for all $z$.*

**Lemma 7.** *There exists constants $\rho, \beta, c_1 > 0$ with the following property. For any $\alpha$ satisfying $\alpha \geq c_1(\log^* |F|)^{-0.99}$, any $\rho$-replicable algorithm solving the statistical interior point problem with success probability $1 - \beta$ on a set $F$ has sample complexity $\widetilde{\Omega}(\log^* |F|)$, even if the underlying distribution is ensured to be $\alpha$-light.*

We next state the following lemma that lower bounds the expectation of the maximum of copies of a distribution under certain conditions.

**Lemma 8.** *Let $X := (1/\varepsilon)^Z$ where $\varepsilon \in (0, 1/2)$ and $Z$ is an $\alpha$-light distribution over the set $F = \{1, \ldots, |F|\}$. Let $X_1, \ldots, X_M$ be i.i.d copies of $X$ where $M \leq 1/(2\alpha)$. Defining $Y = \max\{X_1, \ldots, X_M\}$ we have*

$$\mathbb{E}[X] \leq O(1/M + \varepsilon)\mathbb{E}[Y].$$

We can further generalize the above lemma to $N \geq M$ as follows.

**Lemma 9.** *Let $X = (1/\varepsilon)^Z$ where $Z$ is an $\alpha$-light distribution and $\varepsilon \in (0, 1/2)$. Assume that $M \geq 1/\alpha$ and $N \geq M$. Let $X'$ be a distribution that is $0$ with probability $1 - M/N$ and is sampled from $X$ with probability $M/N$. Let $(a, b) = (\min(Z), \max(Z))$. There exists a constant $c$ such that if $\tau \notin [(1/\varepsilon)^a, (1/\varepsilon)^b]$, then for any $\kappa \in [0, 1]$ the pair $(\tau, \kappa)$ is not $(c(\alpha + \varepsilon))$-competitive for the online pricing instance $X'_1, \ldots, X'_N$.*

As we show in the Appendix, the above lemma combined with the hardness of interior point implies Theorem 3.

**Remark 10.** *The same argument extends to the RD problem. In the proof of Lemma 9, if the price lies outside the range, the algorithm simply accepts the first element $X_i$. Analogously, we may assume that $x(\omega_i)$ and $y(\omega_i)$ are sampled independently. In this case, when the price is not within range, the algorithm selects the "first" element according to the order induced by $y(\omega_i)$. Since $x(\cdot)$ and $y(\cdot)$ are independent, the proof carries over.*

# 5 Replicable Heavy Hitter

In this section, we prove Theorem 5 Let $D$ denote the input distribution. We first present our algorithm to solve the problem. We then move on to the analysis of the replicability of the algorithm and its correctness.

## 5.1 Algorithm

We first take a sample $S^{(1)}$ of size $n_1 = \Theta(\nu^{-1} \log(\rho^{-1} + \beta^{-1} + \nu^{-1}))$ from the distribution. Next, we sample a set $S^{(2)}$ with size $n_2 = \Theta(\log(\frac{n_1}{\min(\rho, \beta)})\nu^{-1} \log^2(\nu^{-1})/\rho^2) \leq \Theta(\log(\nu^{-1})^3 \log(\rho^{-1} + \beta^{-1})\rho^{-2}\nu^{-1})$ samples from the distribution and record, for each element $x \in S^{(1)}$, the number of times it appears in $S^{(2)}$. Sample $\nu'$ uniformly at random from the range $[3/2\nu, 2\nu]$. Let $Y$ denote the set of all elements in $S^{(1)}$ that appear more than $\nu' n_2$ times in $S^{(2)}$, where we remove repetitions of an element so that each element appears at most once in $Y$. If $Y$ is non-empty, we choose an element from it uniformly at random and return as output.

The way we choose the uniform element from $Y$ is important for ensuring replicability. If runtime is not a concern and only sample complexity matters, then one could sort all elements in $\mathcal{X}$ using the shared random bits and pick the element in $Y$ that appears first in the permutation. However, since running time scales linearly with $\mathcal{X}$, this approach is not feasible for large domain sizes. Assuming $r$ denotes the random bits of the algorithm, we will choose a value $\eta(r, x)$ for each $x \in Y$ that depends only on $x$ and $r$ such that, for random $r$, the values $\eta(r, x)$ for $x \in Y$ are all independent and (essentially) distributed uniformly in $[0, 1]$. Importantly, $\eta(r, X)$ does not depend on the samples $S^{(1)}$. We discuss below how this can be achieved using Reed Solomon codes. We then output the element in $Y$ with the lowest value of $\eta(r, .)$.

Intuitively, $\eta(r, x)$ denotes an order of the elements; we emphasize however that this order is unrelated to any pre-existing order among the elements of $\mathcal{X}$ and is chosen uniformly at random. Indeed, it is clear from the definition of the heavy hitter problem that the set $\mathcal{X}$ does not need to be ordered. Throughout the proofs, we will often implicitly assume that the elements are ordered (increasingly) based on $\eta(r, .)$; e.g., we will refer to the element in a set with the smallest value of $\eta(.)$ as its "first" element. The above procedure ensures that if we fix the random bits $r$, any fixed subset of elements will always be in a fixed order as long as the entire set appears in $Y$. This will be important for replicability as it allows us to output the same element regardless of the value of $S^{(1)}$. We omit the dependence of $\eta(., .)$ on $r$ when it is clear from context.

To sample $\eta(r, x)$, we proceed as follows. Let $\mathcal{X}$ denote the ground set of the distribution. Choose a finite field $\mathbb{F}$ such that the size of the field is a power of 2 larger than $|\mathcal{X}|$. Embed each element $x \in \mathcal{X}$ as an element in $\mathbb{F}$. We sample $u_0, \ldots, u_{n_1 - 1}$ uniformly at random from $\mathbb{F}$ and define $\eta(x) = \sum_{i=0}^{n_1 - 1} u_i x^i$. It is clear that $\eta(x)$ depends only on the random bits of the algorithm. It is well-known however that for any fixed values $x_1, \ldots, x_{n_1}$, the values $\eta(x_1), \ldots, \eta(x_{n-1})$ are independent and uniformly random in $\mathbb{F}$. We repeat the process $\Theta(\log(n_1 \max(\beta^{-1}, \rho^{-1})))$ and concatenate the different values of $\eta(x)$ to form a single value $\eta(x)$. The concatenation is made to ensure that, w.h.p, there is no tie among the elements of $Y$ when comparing (see the analysis below).

## 5.2 Analysis

We next state the proof. Due to space constraints, we defer parts of the proofs to the Appendix. Let $p(x) = \Pr_D[x]$ denote the probability of sampling $x$ under the distribution $D$. Let $X^{\geq \nu} = \{x : p(x) \geq \nu\}$ denote all elements for which the sampling probability is more than $\nu$. Note that $X^{\geq \nu}$ has size at most $\frac{1}{\nu}$ by definition. Let $x_{\eta, \nu'} = \arg\min_{x : p(x) \geq \nu'} \eta(x)$ denote the first element with probability larger than $\nu'$. If no such element exists, we set $x_{\eta, \nu'}$ to be Null. We will assume ties are broken in lexicographic order; our analysis will actually condition on the event that ties do not occur which holds with high probability. We note that the value of $x_{\eta, \nu'}$ does not depend on any of the samples and only depends on the input distribution and the random values $\eta(.)$. We will show that, with probability at least $1 - \rho$, the algorithm outputs $x_{\eta, \nu'}$.

Let $X_{\eta}^{\geq \nu} \subseteq X^{\geq \nu}$ denote all of the elements in $S$ that have a higher $p(.)$ value than all elements with smaller $\eta$ values:

$$X_{\eta}^{\geq \nu} = \left\{ x \in X^{\geq \nu} : p(y) < p(x) \text{ for all } y \in X^{\geq \nu} \text{ such that } \eta(y) < \eta(x) \right\},$$

We next define a few events which we later show hold with probability at least $1 - O(\rho)$. We say $\text{Ev}_1$ holds if all elements in $X^{\geq \nu}$ appear in $S^{(1)}$; i.e., $\text{Ev}_1 = \left\{ X^{\geq \nu} \subseteq S^{(1)} \right\}$. For an element $x \in S^{(1)}$, let $\hat{p}(x)$ denote the estimated probability of $x$ based on its repetitions in $S^{(2)}$. Set $\epsilon = \frac{\rho \nu}{10(\ln(\nu^{-1}) + 1)}$. We say $\text{Ev}_2(x)$ holds for an $x$ if either $p(x) < 3\nu$ and $|p(x) - \hat{p}(x)| \leq \epsilon$ or $p(x) \geq 3\nu$ and $\hat{p}(x) \geq 2\nu$. We say $\text{Ev}_2$ holds if $\text{Ev}_2(x)$ holds for all $x \in S^{(1)}$. Let the set $U$ denote the union of the $\epsilon$-neighborhoods of all $x \in X_{\eta}^{\geq \nu}$; formally,

$$U = \cup_{x \in X_{\eta}^{\geq \nu}} (x - \epsilon, x + \epsilon),$$

We say $\text{Ev}_3$ holds if $\nu' \notin U$. As we show in our proofs, we can essentially bound the number of elements in $X_{\eta}^{\geq \nu}$, thereby lower bounding the probability of $\text{Ev}_3$. For an event $\text{Ev}$, let $\neg \text{Ev}$ denote its complementary event.

The following lemmas bound the failure probability of the events.

**Lemma 11.** $\Pr\left[\neg Ev_1\right] \le \min(\rho, \beta)/10$.

**Lemma 12.** $\Pr\left[\neg Ev_2\right] \le \min(\rho, \beta)/10$.

**Lemma 13.** $\Pr\left[\neg Ev_3\right] \le \rho/2$.

We say $Ev_4$ holds if $\eta(x) \ne \eta(y)$ for all distinct $x, y \in S^{(1)} \cup X^{\ge \nu}$. Since $\left|X^{\ge \nu}\right| \le n_1$, we have $\left|S^{(1)} \cup X^{\ge \nu}\right| \le O(n_1)$. It follows that

$$\Pr\left[\neg Ev_4\right] \le O(n_1^2) 2^{-\Theta(\log(n_1 + \beta^{-1} + \rho^{-1}))} \le \frac{\min(\rho, \beta)}{20},$$

where for the second inequality we have assumed that the hidden constant under $\Theta(.)$ in the exponent is large enough.

**Lemma 14.** *Assume that $Ev_1, Ev_2, Ev_3$ and $Ev_4$ hold. The output of the algorithm is $x_{\eta, \nu'}$.*

Now, we procceed with the proof of Theorem 5.

*Proof of Theorem 5.* It is clear that the algorithm has the desired sample complexity. For replicability, Putting together the above lemmas, we conclude that the probability that the algorithm outputs $x_{\eta, \nu'}$ is at least $1 - \rho$. It follows that two independent runs of the algorithm with shared randomness will have the same output with probability at least $1 - 2\rho$, implying that the algorithm is $(2\rho)$-replicable. It is clear that by replacing $\rho$ with $\rho/2$, the algorithm becomes $\rho$-replicable with the same sample complexity up to constant factors.

We next prove correctness. We show that assuming $Ev_1, Ev_2, Ev_4$ hold, the algorithm's output is correct; i.e., **(a)** it is either `Null` or a $\nu$-heavy hitter and **(b)** if a $(4\nu)$-heavy hitter exists, it is not `Null`. For proving **(a)**, observe that if the output is not `Null` then it is some $x \in S^{(1)}$ such that $\hat{p}(x) > \nu'$. By $Ev_2$, this implies that $p(x) \ge \nu' - \epsilon \ge \nu$, and we are done. For **(b)**, letting $x^*$ denote the point satisfying $p(x^*) \ge 4\nu$, we have $x^* \in S^{(1)}$ because of $Ev_1$ and $\hat{p}(x^*) \ge 2\nu$ by $Ev_2$. It follows that the output will not be `Null` and the proof is complete.

$\square$

# 6 Conclusion

In this work, we investigated replicability in online decision-making, with a focus on the Replicable Online Pricing (ROP) problem and its sample complexity. By designing efficient replicable algorithms and establishing fundamental lower bounds, we highlighted the trade-offs between replicability, competitiveness, and sample efficiency. We also explored the broader implications of our results for the delegation problem, a key economic model of principal-agent interactions, analyzing strategies to align incentives while maintaining decision-making autonomy. As part of our technique, we obtained a new algorithm for the replicable heavy hitter problem which may be of independent interest. Our findings enhance the understanding of replicability in both algorithmic and economic contexts, underscoring its importance in ensuring fairness, transparency, and reliability.

An interesting direction for future work is applying our techniques to study replicability in other pricing problems. Another promising direction is to extend our methods, particularly the lower bounds, to study broader notions of robustness—for example, allowing replicable algorithms to produce approximately equal outputs. Additionally, while we focused on multiplicative guarantees to align with the standard competitive analysis framework, it would be interesting to explore algorithms that allow additive error. Improving the dependence of our bounds on any of the relevant parameters is also an interesting direction for future work. Finally, it would be interesting to analyze the practical implications of work using empirical evaluations.

# 7 Acknowledgements

The work is partially supported by DARPA QuICC, ONR MURI 2024 award on Algorithms, Learning, and Game Theory, Army-Research Laboratory (ARL) grant W911NF2410052, NSF AF:Small grants 2218678, 2114269, 2347322, and Royal Society grant IES\R2\222170.

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

# A Further related work

## A.1 Replicability

The concept of algorithmic replicability and reproducibility has attracted considerable attention in recent years, leading to the development of various theoretical frameworks and practical algorithms across multiple domains. [ILPS22] introduced the notion of reproducible algorithms within the context of learning, highlighting the balance between randomness, accuracy, and reproducibility. They developed a theoretical foundation for reproducible algorithms and examined reproducibility in statistical query (SQ) algorithms, approximate heavy hitters, medians, and learning halfspaces. They also explored some inherent trade-offs and lower bounds associated with reproducibility.

Building on this foundation, replicability has been examined across several fields. For instance, [EKK+23] extended the concept to stochastic bandits and developed replicable policies that could achieve optimal regret bounds comparable to their non-replicable counterparts. [EHKS23] initiated the study of replicable RL algorithms, providing provably replicable versions of parallel value iteration and R-Max in episodic settings. Similarly, [KVYZ23] investigated replicability within reinforcement learning (RL), focusing on discounted tabular Markov Decision Processes (MDPs) with access to a generative model. They provided replicable algorithms for $(\epsilon, \delta)$-optimal policy estimation. They also explored TV indistinguishability as a relaxed form of replicability and introduced the notion of approximate replicability.

[EKM+23] proposed replicable algorithms for statistical clustering by applying replicability concepts to problems like $k$-medians, $k$-means, and $k$-centers. The computational and statistical costs of replicability in high-dimensional tasks, such as multi-hypothesis testing and mean estimation, were examined by [HIK+24a]. Their study established an equivalence between optimal replicable algorithms and high-dimensional isoperimetric tilings, resolving open problems related to sample complexity.

[LY24] addressed replicability in uniformity testing, presenting a replicable uniformity tester using only $O\left(n\varepsilon^{-2}\rho^{-1}\right)$ samples, and providing a nearly matching sample complexity lower bounds for replicable uniformity testing of symmetric algorithms invariant under domain relabeling. In convex optimization, [ZYKH23] demonstrated that optimal reproducibility and near-optimal convergence could be simultaneously achieved in various optimization settings. Additionally, [KKL+24] studied replicable algorithms for learning large-margin halfspaces, presenting dimension-independent algorithms with improved sample complexity.

In another line of work, the relation between replicability and other stability notions was explored by [BGH+23]. They provided algorithmic reductions between replicability, perfect generalization, and approximate differential privacy, while also identifying computational separations that underline fundamental differences between these stability concepts. [KKMV23] explored Total Variation (TV) indistinguishability as a measure of learning rule stability and investigated information-theoretic equivalences between TV indistinguishability, replicability, and differential privacy. Lastly, [KKVZ24] focused on the computational connections between replicability and learning paradigms like online learning, private learning, and SQ learning.

**Replicable mean and median estimation.** Replicable algorithms for mean and median estimation were first developed by Impagliazzo, Lei, Pitassi, and Sorrell [ILPS22], who introduced general-purpose techniques for statistical queries under replicability constraints. Their replicable median algorithm has sample complexity exponential in $\log^* |X|$. Bun, Gaboardi, Hopkins, Impagliazzo, Lei, Pitassi, Sivakumar, and Sorrell [BGH+23] later improved on this algorithm by drawing connections to differential privacy, obtaining an algorithm with sample complexity that is polynomial in $\log^* |X|$, albeit without computational efficiency.

**Replicable heavy hitters.** Prior work has also studied replicable algorithms for heavy hitter identification. Impagliazzo et al. [ILPS22] and Esfandiari et al. [EKM+23] provide algorithms that return a list of candidate heavy hitters, with sample complexity scaling cubically in $1/\nu$. More recently, Hopkins et al. [HIK+24a] improved the dependence on the failure and replicability parameters for expected sample complexity, but still maintained a $\nu^{-3}$ dependence. In contrast, we consider a more specialized variant where only a single heavy hitter needs to be output (assuming one exists), and

give a new algorithm with nearly linear $\nu^{-1}$ dependence, which is necessary for our lower bound construction in Theorem 3.

## A.2 Prophet Inequalities and delegation

The study of prophet inequalities and optimal stopping theory has seen significant growth over the years, particularly in mechanism design and online optimization. Originally introduced by [KS77], prophet inequalities provide a framework for comparing the expected outcomes of a prophet, who knows future events, to those of a decision-maker without such foresight. Kennedy [Ken87] extended the framework to multi-choice optimal stopping, identifying the best possible universal constants for scenarios where multiple selections are allowed. [AGSC02] further refined the prophet inequality framework by introducing ratio prophet inequalities in settings where several stopping rules are available to the decision-maker and offering recursively evaluated constants that provide tight bounds for these scenarios.

The integration of prophet inequalities into mechanism design has also seen significant advancements. [HKS07] applied these concepts to online mechanism design, developing approximately efficient automated systems through new prophet inequalities motivated by the auction setting. [FGL15] investigated combinatorial auctions with posted price mechanisms in a Bayesian setting and developed the first constant-factor DSIC mechanism for Bayesian submodular combinatorial auctions.

[KW12] generalized prophet inequalities to matroid settings, providing tight bounds under matroid constraints and highlighting applications in Bayesian mechanism design. Further generalizations include the work of [AHL12], who introduced online prophet-inequality matching in bipartite graphs, particularly relevant to ad allocation, and [AHL13], who explored stochastic generalized assignment problems within this framework.

There is also another line of work studying prophet inequalities on i.i.d. distributions, starting with the work of [HK82]. Many years later, [AEE$^+$17] presented a threshold-based algorithm that surpassed the theoretical bound of $1 - 1/e$ previously conjectured by [HK82]. Then, [CFH$^+$17] investigated the performance of posted price mechanisms when customers arrive in an unknown random order, considering both adaptive and nonadaptive cases. They established a tight bound of $1 - 1/e$ for the nonadaptive case, even when valuations are i.i.d., and proposed a mechanism achieving a better approximation factor of $0.745$ in the adaptive case with i.i.d. valuations. More recently, [CDFS19] studied the i.i.d case of prophet inequality when the underlying distribution is unknown. They showed that, without distributional knowledge, the best achievable bound is $1 - 1/e$, aligning with the classic secretary problem. However, they further demonstrated that access to a limited number of samples from the distribution could significantly improve performance.

It is worth noting that many (though not all) works in the prophet inequalities literature use (possibly dynamic) pricing based algorithms to obtain their results and that any algorithm for prophet inequalities admits a pricing-based implementation [BHK$^+$24].

**Sample-based prophet inequalities.** The sample-based study of prophet inequalities—where the algorithm has access only to samples from the underlying distributions—has become an active and evolving area of research. This direction was initiated by Azar, Kleinberg, and Weinberg [AKW14], who drew connections between this setting and the classical secretary problem. In the case of fixed arrival order, Rubinstein, Wang, and Weinberg [RWW19] showed that even a single sample per distribution suffices to achieve the optimal 1/2-approximation. In the i.i.d. variant, they also proved that a constant number of samples is enough, with later work by Correa et al. [CCES24] refining the required sample complexity. When the arrival order is random, as in the prophet secretary model, Correa et al. [CCES22] established that a single sample still enables a 0.635-approximation. This sampling approach has since been extended to more structured selection problems, such as those involving matroids and matchings [CDF$^+$22, KNR22]. More recently, Cristi and Ziliotto [CZ24] developed a unified argument showing that constant sample access suffices to achieve approximate optimality across both prophet-secretary and free-order settings, despite the exact optimal approximation ratios for these variants remaining open.

**Delegation** The concept of delegation, a core topic in economic theory since its introduction by [Hol80], has been explored in many different ways over the years. However, [KK18] were the first to analyze the efficiency loss in delegated search, proposing methodologies to bound this loss and

demonstrating how simple threshold-based mechanisms can approximate the efficiency of direct search. Their work established a connection between delegation and prophet inequalities, resulting in a surge of attention to this problem.

[BD21] introduced the concept of delegated stochastic probing, merging delegation with stochastic probing problems. [HMRS24] explored the online learning aspects of delegation through a repeated delegated choice model. Their work extends the delegation framework by incorporating dynamic learning of solution distributions, focusing on minimizing cumulative regret in various strategic settings. [STCR23] address the delegation of classification tasks to rational agents. They propose incentive-aware frameworks that use performance-based contracts to align agents' interests with those of the principal, connecting contract design with statistical hypothesis testing. [SRH23] investigate mechanisms in multi-agent settings without monetary incentives. [BHHS24] delve into delegated online search, where agents sequentially inspect options and propose solutions in real time. [BHKS25] show that the correspondence between delegated choice and prophet inequalities holds if and only if the feasibility constraint forms a matroid. They also establish a separation between the two problems, proving that, unlike prohpet inequalities, delegated choice admits a constant-factor approximation under downward-closed constraints.

# B   Proofs in Section 3

## B.1   Proof of Lemma 4

*Proof of Lemma 4.* Sample $B \sim \text{Bernoulli}(\kappa)$. Define $\text{OPT} = \max_i X_i$. We need to show that,

$$\mathbb{E}\left[X_{i(\tau,B)}\right] \geq (1/2 - \gamma)\mathbb{E}\left[\text{OPT}\right],$$

where we set $X_{i(\tau,B)} = 0$ when $i(\tau,B) = \infty$. We can rewrie the left hand side as

$$
\begin{aligned}
& \mathbb{E}\left[X_{i(\tau,B)}\right] \\
&= \sum_{i=1}^{N} \mathbb{E}\left[X_i \mathbb{1}\left\{i(\tau,B) = i\right\}\right] \\
&= \sum_{i=1}^{N} \mathbb{E}\left[(\tau + (X_i - \tau))\mathbb{1}\left\{i(\tau,B) = i\right\}\right] \\
&= \sum_{i=1}^{N} \mathbb{E}\left[\tau\mathbb{1}\left\{i(\tau,B) = i\right\}\right] + \sum_{i=1}^{N} \mathbb{E}\left[(X_i - \tau)\mathbb{1}\left\{i(\tau,B) = i\right\}\right]
\end{aligned}
$$

The first term can be lower bounded as

$$\tau \sum_{i=1}^{N} \mathbb{E}\left[\mathbb{1}\left\{i(\tau,B)\right\} = i\right] = \tau\Pr\left[i(\tau,B) \neq \infty\right]$$

As for the second term, define $Z_i \in \{0,1\}$ to be 1 if $X_i \geq \tau$ and $B = 0$ or $X_i > \tau$ and $B = 1$. Observe that

$$\mathbb{1}\left\{i(\tau,B) = i\right\} \geq Z_i \prod_{j \neq i}(1 - Z_j).$$

This is because if $Z_i = 1$ and all other $Z_j$ are 0, then we must have $i(\tau, B) = i$. We can therefore obtain the following lower bound

$$
\mathbb{E}\left[(X_i - \tau)\mathbb{1}\left\{i(\tau, B) = i\right\}\right] \geq \mathbb{E}\left[(X_i - \tau)Z_i\prod_{j \neq i}(1 - Z_j)\right]
$$

$$
\geq \mathbb{E}\left[(X_i - \tau)Z_i\right]\mathbb{E}\left[\prod_{j \neq i}(1 - Z_j)\right]
$$

$$
\geq \mathbb{E}\left[(X_i - \tau)Z_i\right]\mathbb{E}\left[\prod_{j=1}^{N}(1 - Z_j)\right]
$$

$$
= \mathbb{E}\left[(X_i - \tau)Z_i\right]\Pr\left[i(\tau, B) = \infty\right]
$$

where for the first inequality we have used the fact that $X_1, \ldots, X_N$ are independent. We note however that $(X_i - \tau)Z_i = [X_i - \tau]^+$. This follows from a case analysis. When $X_i < \tau$, then we have $Z_i = 0$ and both sides are 0. When $X_i = \tau$, then again both sides are 0. When $X_i > \tau$, we have $Z_i = 1$ and both sides are equal to $X_i - \tau$. It follows that

$$
\sum_{i=1}^{N}\mathbb{E}\left[(X_i - \tau)\mathbb{1}\left\{i(\tau, B) = i\right\}\right] \geq \Pr\left[i(\tau, B) = \infty\right]\sum_{i=1}^{N}[X_i - \tau]^+
$$

Observe however that

$$
\Pr\left[i(\tau, B) \neq \infty\right] = (1 - \kappa)\Pr\left[\left(\max_i X_i\right) \geq \tau\right] + \kappa\Pr\left[\left(\max_i X_i\right) > \tau\right].
$$

Since $(\tau, \gamma)$ is a $\gamma$-approximate median pair for $\max_i X_i$, the above probability is in the range $(1/2 - \gamma, 1/2 + \gamma)$. It follows that

$$
\Pr\left[i(\tau, B) \neq \infty\right], \Pr\left[i(\tau, B) = \infty\right] \geq 1/2 - \gamma.
$$

This in turn implies

$$
\mathbb{E}\left[X_{i(\tau, B)}\right] \geq (1/2 - \gamma)\mathbb{E}\left[\left(\tau + \sum_{i=1}^{N}[X_i - \tau]^+\right)\right]
$$

$$
\geq \mathbb{E}\left[\left(\tau + \max_{i=1}^{N}[X_i - \tau]^+\right)\right]
$$

$$
\geq \mathbb{E}\left[\left(\max_{i=1}^{N} X_i\right)\right]
$$

$$
= (1/2 - \gamma)\text{OPT}.
$$

$\square$

## B.2 Application to delegation

In this section we prove Theorem 2, which we first restate for completeness:

**Theorem 2.** *Assume that the distribution $\mathcal{D}$ is supported over a known finite set $\mathcal{X}$. For any $\epsilon \in (0, 1/2)$ and $\rho, \beta > 0$, there exists an algorithm solving the RD problem with parameters $(1/2 - \epsilon, \rho, \beta)$ using at most $N \cdot \text{poly}\left(\log^* |\mathcal{X}|, \rho^{-1}, \beta^{-1}, \epsilon^{-1}\right)$ samples from $\mathcal{D}$.*

*Proof.* We know that for any threshold $\tau$ chosen by the principal, the best strategy for the agent would be to choose $\omega_i$ with the highest corresponding value $y(\omega_i)$ among all those that meet the threshold $x(\omega_i) \geq \tau$ or $x(\omega_i) > \tau$ depending on the value of $B$. We will denote this as $\omega^{(\tau, B)}$ henceforth. Note that $\omega^{(\tau, B)}$ might not exist, in which case neither the principal nor the agent would gain anything regardless of the agent's proposal. However, the existence of $\omega^{(\tau, B)}$ warrants the principal's approval and a gain of $x(\omega^{(\tau, B)})$. As $\omega^{(\tau, B)}$ can also be seen as the first task with either $x(\omega) \geq \tau$ or $x(\omega) > \tau$ when the tasks are sorted based on their corresponding value $y$ in descending order, the problem of

choosing an appropriate threshold $\tau$ for the principal is closely related to choosing an appropriate threshold for the prophet pricing problem.

We propose that the principal draw $N \cdot s$ samples from $\mathcal{D}$, denoted by $\omega'_{i,j}$ for each $1 \leq i \leq N$ and $1 \leq j \leq s$, where $s$ denotes the sample complexity of Proposition 3. For each $1 \leq j \leq s$, we then define $x_j := \max_{1 \leq i \leq N} x(\omega'_{i,j})$. This construction yields $x_1, \ldots, x_s$ as $s$ independent samples from the random variable $X$, where $X := \max_{1 \leq i \leq N} x(\omega_i)$.

Applying the given $\rho$-replicable algorithm for computing a $\gamma$-approximate median pair $(\tau, \kappa)$ on the samples $x_1, \ldots, x_s$, we obtain a $\rho$-replicable $\gamma$-approximate median pair $(\gamma, \kappa)$ for $D$. We report this as the principal's mechanism, and by Lemma 15 $\tau$ would be $(1/2 - \gamma)$ competitive, which hence completes the proof. $\qquad\square$

**Lemma 15.** *Let $(\tau, \kappa)$ be a $\gamma$-approximate median for the distribution of $\max_{1 \leq i \leq N} x(\omega_i)$. Then, $(\tau, \kappa)$ would yield a $(1/2 - \gamma)$-competitive utility for the principal.*

*Proof.* The proof follows the same structure as Lemma 4. Specifically, define $X_i = x(\omega_i)$. Sample $B \sim \text{Bernoulli}(\kappa)$. Define OPT $= \max_i X_i$. As before, we need to show that

$$\mathbb{E}\left[X_{i(\tau,B)}\right] \geq (1/2 - \gamma)\mathbb{E}\left[\text{OPT}\right],$$

where we set $X_{i(\tau,B)} = 0$ when $i(\tau, B) = \infty$. Note that the definition of $i(\tau, B)$ is different from that of Lemma 4; instead of picking the first element in the order of time, we are picking the first element in the order specified by $y(\omega_i)$. Just as before however, the algorithms obtains $\tau$ with probability $\Pr\left[i(\tau, B) \neq \infty\right]$ and $\sum_i \mathbb{E}\left[[X_i - \tau]^+\right]$ with probability $\Pr\left[i(\tau, B) = \infty\right]$. As before, the probabilities can be lower bounded with $1/2 - \gamma$ and the proof goes through. $\qquad\square$

## C    Proofs in Section 4

We first state the following standard tools from probability theory that will be used in the analysis.

**Lemma 16.** *Let $X$ be a distribution over the set $\mathcal{X} = \left\{x_1, \ldots, x_{|\mathcal{X}|}\right\} \subseteq \mathbb{R}^{\geq 0}$ where $x_i < x_{i+1}$ for $i \in |\mathcal{X}| - 1$. Let $x_0 = 0$. The expectation of $X$ satisfies the following equality:*

$$\mathbb{E}\left[X\right] = \sum_{i=1}^{|\mathcal{X}|} (x_i - x_{i-1}) \Pr\left[X \geq x_i\right].$$

**Lemma 17** (Lemma A.5. in [BHSS23]). *Let $X_1, \ldots, X_n$ be i.i.d random variables and let $\tau$ be an arbitrary value. Let $p$ denote the probability $\Pr\left[X_i \geq \tau\right]$. Then $\Pr\left[\max_i X_i \geq \tau\right] \geq \frac{1}{2}(\min\left\{np, 1\right\})$.*

We note that the lower bound in the above lemma can be written as $\Omega(np)$ when $p \leq 1/n$.

*Proof of Lemma 7.* Let $\rho, \beta$ be half of the corresponding constants from Proposition 5. Let $A$ be a $\rho$-replicable algorithm with sample complexity $n$ that can solve the problem with failure probability at most $\beta$ for $\alpha$-light distributions. We transform this into an algorithm $A'$ for all (not necessarily light) distributions as follows. First, we invoke the $\rho$-replicable $\nu$-heavy hitter algorithm with failure probability $\beta$ from Theorem 5 for $\nu = \alpha/4$ and $\gamma = 4$ If the algorithm's output is not Null, we return this as the output. Otherwise, we invoke algorithm $A$ and return its output. We show below that $A'$ is $(2\rho)$-replicable and has failure probability at most $2\beta$. By Proposition 5, this implies a lower bound on the sample complexity of $A'$, which in turn implies a lower bound on the sample complexity of $A$.

We next analyze the algorithm $A'$.

**Correctness.**    Assume that the distribution is not $\alpha$-light. Then the heavy hitter algorithm will return some value $z \neq$ Null such that $\Pr\left[Z = z\right] > \nu > 0$ with probability $1 - \beta$. Therefore, the output will be correct. If the distribution is $\alpha$-light, the heavy hitter algorithm will output either a value $z \neq$ Null such that $\Pr\left[Z = z\right] > 0$, or it will output Null. If the output is Null, then algorithm $A$ will output a correct answer with probability at least $1 - \beta$. A union bound implies that the answer is correct with probability at least $1 - 2\beta$ in this case as well.

**Replicability.** Consider two runs with samples $S_1$ and $S_2$ and the same random bits. With probability at least $1 - \rho$, the heavy hitter algorithm will produce the same output for the two runs. If the output is not `Null`, then we will obtain the same answers. Otherwise, we will invoke algorithm $A$ in both cases, which will produce the same answer with probability $1 - \rho$. Taking union bound implies that the algorithm is $(2\rho)$-replicable.

**Sample complexity.** The algorithm requires $n$ samples for the potential run of algorithm $A'$ by assumption and requires at most $O(\alpha^{-1} \log^3(1/\alpha))$ samples for the single heavy hitter algorithm by Theorem 5. This means that the overall sample complexity is $n + O(\alpha^{-1} \log^3(1/\alpha))$. Since the algorithm is $(2\rho)$-replicable and has failure probability at most $2\beta$, Proposition 5 implies that

$$n + O(\alpha^{-1} \log^3(1/\alpha)) \geq \tilde{\Omega}(\log^* |F|).$$

Given the assumption on $\alpha$, if $c_1$ is small enough, this means that $n \geq \tilde{\Omega}(\log^* |F|)$ as claimed, finishing the proof. $\qquad\square$

*Proof of Lemma 16.* Evaluating the right hand side,

$$
\begin{aligned}
\sum_{i=1}^{|\mathcal{X}|} (x_i - x_{i-1}) \Pr\left[X \geq x_i\right] &= \sum_{i=1}^{|\mathcal{X}|} \sum_{j \geq i} (x_i - x_{i-1}) \Pr\left[X = x_j\right] \\
&= \sum_{j=1}^{|\mathcal{X}|} \sum_{i \leq j} (x_i - x_{i-1}) \Pr\left[X = x_j\right] && \textit{(Double counting)} \\
&= \sum_{j=1}^{|\mathcal{X}|} (x_j - x_0) \Pr\left[X = x_j\right] && \textit{(Telescoping sum)} \\
&= \mathbb{E}\left[X\right] && \textit{(Definition of expectation)}
\end{aligned}
$$

$\qquad\square$

*Proof of Lemma 8.* Set $x_i = (1/\varepsilon)^i$ for $i \in [|F|]$. Choose smallest $i$ such that $\Pr\left[X \geq x_i\right] \leq \frac{1}{M}$. By definition of $i$ we have $\Pr\left[X \geq x_{i-1}\right] \geq 1/M$. Since the distribution $Z$ (and therefore $X$) is $\alpha$-light, we must have $\Pr\left[X = x_{i-1}\right] \leq \alpha \leq 1/(2M)$, which in turn implies $\Pr\left[X \geq x_i\right] \geq 1/(2M)$. Since $\Pr\left[X \geq x_j\right]$ is decreasing in $j$, this means that $\Pr\left[X \geq x_j\right] \leq 1/M$ for $j \geq i$. We can lower bound the expectation of $Y$ in two ways. Firstly,

$$
\begin{aligned}
\mathbb{E}\left[Y\right] &= \sum_j (x_j - x_{j-1}) \Pr\left[Y \geq x_j\right] && \textit{(Lemma 16)} \\
&\geq \sum_{j \geq i} (x_j - x_{j-1}) \Pr\left[Y \geq x_j\right] && \textit{(Since $x_j > x_{j-1}$)} \\
&\geq \Omega(M) \sum_{j \geq i} (x_j - x_{j-1}) \Pr\left[X \geq x_j\right] && \textit{(Lemma 17 and $\Pr\left[X \geq x_j\right] \leq 1/(M)$),}
\end{aligned}
$$

Secondly, since $x_j - x_{j-1} \geq x_j/2$ (because $\varepsilon < 1/2$) and $\sum_{j=1}^{j'}(x_j - x_{j-1}) \leq x_{j'+1}$ for all $j, j'$, we obtain

$$
\begin{aligned}
\mathbb{E}[Y] &= \sum_j (x_j - x_{j-1})\Pr[Y \geq x_j] && \text{\textit{(Lemma 16)}} \\
&\geq (x_i - x_{i-1})\Pr[Y \geq x_i] && \text{\textit{(Since }} x_j > x_{j-1}) \\
&\geq \Theta(1)(x_i - x_{i-1}) && \text{\textit{(Lemma 17 and }} \Pr[X \geq x_i] \geq 1/(2M)) \\
&\geq \Theta(1)x_i && \text{\textit{(Since }} \varepsilon < 1/2) \\
&\geq \Theta(\varepsilon^{-1})(x_{i-1} - x_0) && \text{\textit{(Since }} x_i = \varepsilon^{-1}x_i \text{ \textit{and} } x_0 = 0) \\
&= \Theta(\varepsilon^{-1})\sum_{j=1}^{i-1}(x_j - x_{j-1}) && \text{\textit{(Telescoping sum)}} \\
&\geq \Theta(\varepsilon^{-1})\sum_{j=1}^{i-1}(x_j - x_{j-1})\Pr[X \geq x_j].
\end{aligned}
$$

Combining the above inequalities we obtain

$$
\begin{aligned}
\mathbb{E}[X] &= \sum_j (x_j - x_{j-1})\Pr[X \geq x_j] && \text{\textit{(Lemma 16)}} \\
&\leq \sum_{j<i}(x_j - x_{j-1})\Pr[X \geq x_j] + \sum_{j \geq i}(x_j - x_{j-1})\Pr[X \geq x_j] \\
&\leq O(1/M + \varepsilon)\mathbb{E}[Y],
\end{aligned}
$$

finishing the proof. $\qquad\square$

*Proof of Lemma 9.* Let $C_1, \ldots, C_N$ be i.i.d Bernoulli variables such that $\Pr[C_i = 1] = M/N$. Sampling $X_1, \ldots, X_N$ i.i.d from the distribution of $X$, we can assume that $X_i' = C_i X_i$. If the price $\tau$ is not in the range $[(1/\varepsilon)^a, (1/\varepsilon)^b]$, there are two possibilities. The first is that it is $> (1/\varepsilon)^b$. In this case, accepting the first element above $\tau$ is equivalent to not accepting anything, which means $\tau$ is only 0-competitive. We therefore focus on the case $\tau < (1/\varepsilon)^a$. We further assume $\tau > 0$ since clearly a non-zero price is better than 0. If $\tau \in (0, (1/\varepsilon)^a)$, then accepting the first element above or equal to $\tau$ or accepting the first element strictly greater than $\tau$ are both equivalent to accepting the first element $X_i'$ such that $C_i = 1$, assuming such an element exists. Since $C_i$ and $X_i$ are independent, this means that the expected value of the algorithm is at most $\mathbb{E}[X_1]$, where we note that we have an inequality because with positive probability all $C_i$ are 0. Define $Y = \max\{X_1', \ldots, X_M'\}$. We need to show that $\mathbb{E}[Y] \geq \Omega(\frac{1}{\alpha+\varepsilon})\mathbb{E}[X_1]$. Since each $C_i$ is 1 with probability $M/N$, we have $\mathbb{E}\left[\sum_{i=1}^N C_i\right] = M$. By Chernoff, this implies

$$
\Pr\left[\sum_{i=1}^N C_i \leq M/2\right] \leq \left(\frac{e^{-1/2}}{(1/2)^{1/2}}\right)^M < 1 - \Omega(1).
$$

Therefore, with probability $\Omega(1)$, we will have $\sum_{i=1}^N C_i \geq M/2$. Conditioned on this event, since $X_1, \ldots, X_N$ are i.i.d and independent of $C_i$, $Y$ is a maximum of at least $1/(2\alpha)$ copies of $X_i$. We can assume, without loss of generality, that $\alpha \leq 1/8$. If $\alpha \geq 4$, then setting $c > 8$ the lemma holds trivially as one cannot have a price that is $c$-competitive for $c > 1$. It follows that $\frac{1}{2\alpha} > 4$ which in turn implies $\lfloor\frac{1}{2\alpha}\rfloor \geq \Omega(1/\alpha)$. Therefore, $(\lfloor\frac{1}{2\alpha}\rfloor)^{-1} \leq O(\alpha)$.

Let $Y'$ denote the maximum of the first $\lfloor 1/(2\alpha)\rfloor$ copies of $X$. It is clear that $Y \geq Y'$. By Lemma 8 however,

$$
\mathbb{E}[X_1] \leq O\left(\left(\left\lfloor\frac{1}{2\alpha}\right\rfloor\right)^{-1} + \varepsilon\right) \leq O(\alpha + \varepsilon)\mathbb{E}[Y'].
$$

Therefore, conditioned on the event $\sum_{i=1}^N C_i \geq M/2$ we have $\mathbb{E}[Y] \geq \Omega(\frac{1}{\alpha+\varepsilon})\mathbb{E}[X_1]$. Since the event happens with constant probability, this holds without the conditioning as well, finishing the proof. $\qquad\square$

*Proof of Theorem 3.* Let $\rho', \beta', c_1'$ denote the values of the constants from Lemma 7. Let $c'$ denote the value of $c$ in Lemma 9. We assume without loss of generality that $c' \geq 1$ since increasing $c'$ doe not violate the lemma. Set $\rho = \rho'/2, \beta = \beta'/2$ and $c_1 = 4c'c_1'$. Set $\alpha' = \alpha/(4c'), M = \lceil 1/\alpha' \rceil$ and set $c_2 = 8c'$.

Note that this implies

$$N \geq c_2/\alpha = 2/\alpha' \geq M.$$

For the final inequality, we have used the fact that $2/\alpha' \geq \lceil 1/\alpha' \rceil$. This follows from the fact that $1/\alpha' \geq \frac{4c'}{\alpha} \geq 4$; note that $\alpha \leq 1$ as otherwise the lemma holds trivially.

Let $A$ be an algorithm that is $\rho$-replicable and with probability $1 - \beta$, outputs a price pair that is $\alpha$-competitive for $N$ i.i.d variables $X_1', \ldots, X_N'$ and has sample complexity $n$. We need to show that $n \geq \widetilde{\Omega}(\alpha \log^* |\mathcal{X}|)$.

We will build an algorithm $A'$ for the replicable interior point problem on $\alpha'$-light distributions. Let $Z$ be an $\alpha'$-light distribution and set $\varepsilon = \alpha'$. Create the distribution $X'$ as in Lemma 9. Give this as input to the algorithm $A$ and denote its output price pair with $(\tau, \kappa)$. Note that the algorithm will attempt to sample from the distributions of $X_i'$, which in turn requires us to potentially sample from $Z$. The number of samples the algorithm will require from $Z$ is random. If the algorithm $A$ ends up requiring more than $n' = C(\log(\beta^{-1}) + \log(\rho^{-1}))Mn$ samples from $Z$, (where $C$ will be specified later), we halt the algorithm and output `Null`. Otherwise, we round the value $\log_{1/\varepsilon}(\tau)$ to the nearest integer and output it.

**Correctness.** We will show that the algorithm's output is correct with probability at least $1 - \beta$. Assume for simplicity that instead of halting the algorithm if we require more than $n'$ samples, we allow the algorithm to continue and calculate the price $\tau$, but output `Null` in the end. It is clear that this does not change the algorithm's output.

The expected number of samples the algorithm takes from $Z$ is $\frac{M}{N} \cdot N \cdot n = Mn \leq \frac{2}{\alpha'}n$. Therefore, by Chernoff, the probability that the number of samples we require exceeds $n'$ is at most $e^{-\Theta(n')}$, which is $\leq \beta/8$ if we set $C$ to be large enough. Furthermore, with probability at least $1 - \beta$, the output of the algorithm $A$ will be a price pair $(\tau, \kappa)$ that is $\alpha$-competitive. Invoking Lemma 9, we conclude that such a $\tau$ must be in range $[(1/\varepsilon)^a, (1/\varepsilon)^b]$ since otherwise it would be at most $(c(\alpha' + \varepsilon))$-competitive and $c(\alpha' + \varepsilon) = \alpha/2$. Note that the preconditions of the lemma hold. We have $\varepsilon = \alpha' = \alpha/(4c') < 1/2$. We also have $M = \lceil 1/\alpha' \rceil \geq 1/\alpha'$ and $N \geq M$.

If $\tau$ is in the range $[(1/\varepsilon)^a, (1/\varepsilon)^b]$, then the output of $A'$ will be an interior point of $Z$. Note that if $\tau$ does not correspond to a value $\varepsilon^z$ for an integer $z$, it does not matter how we round $\log_{\varepsilon^{-1}}(\tau)$; since $\tau$ is in the range $[\varepsilon^a, \varepsilon^b]$, both options for rounding are acceptable. Therefore, the algorithm $A'$ is correct with probability at least $1 - \beta(1 + 1/8) \geq 1 - \beta'$.

**Replicability.** As before, we assume for simplicity that instead of halting the algorithm if we require more than $n'$ samples, we allow the algorithm to continue and calculate the price pair $(\tau, \kappa)$, but output `Null` in the end. If we run the algorithm $A'$ with two separate samples $S_1, S_2$ and the same random bits, with probability $1 - \rho/4$, neither of the two runs return `Null`. This follows from the fact that the probability of returning `Null` is at most $e^{-\Theta(n')}$ which is $\leq \rho/4$ given a large enough $C$. Additionally, with probability at least $1 - \rho$, both runs of $A$ return the same output. It follows that with probability at least $1 - \rho'$, the output is the same.

**Sample complexity.** By definition of $c'$, we have $\alpha' = \alpha/(4c') \geq c_1'((\log^* |F|)^{-0.99})$. The algorithm $A'$ produces an interior point of $Z$ with probability at least $1 - \beta'$ and is $\rho'$-replicable. It follows from Lemma 7 that its sample complexity $n'$ must be at least $\widetilde{\Omega}(\log^* |F|)$. Since $\rho, \beta$ were constants, we have $n' = O(Mn) = O(n/\alpha)$, where the second inequality follows from the fact that, since $2/\alpha' \geq M$. Therefore, $n/\alpha \geq \widetilde{\Omega}(\log^* |F|)$ and the proof is complete.

$\square$

# D    Proofs in Section 5

*Proof of Lemma 11.* For any $x \in X^{\geq \nu}$, $\Pr\left[x \notin S^{(1)}\right] \leq (1 - p(x))^{n_1} \leq e^{-np_1(x)} \leq e^{-n_1 \nu}$ Since $n_1 = \Theta(v^{-1} \log(v^{-1} + \rho^{-1} + \beta^{-1}))$, with the correct constants under $\Theta(.)$ we have $e^{-n_1 \nu} \leq \nu \min(\rho, \beta)$. Since $\left|X^{\geq \nu}\right| \leq \nu^{-1}$, taking a union bound the claim follows.    $\square$

*Proof of Lemma 12.* We will show that for any $x \in S^{(1)}$ we have $\Pr\left[\neg \text{Ev}_2(x)\right] \leq \frac{\min(\rho, \beta)}{10n_1}$. Taking a union bound completes the proof. Recall that $\hat{p}(.)$ is an estimate of $p(.)$ using $n_2$ samples. Therefore, by Chernoff (Equation (2)),

$$\Pr\left[|\hat{p} - p| \geq \epsilon\right] \leq 2e^{-\frac{n_2 \epsilon^2}{3p}}.$$

If $p \leq 3\nu$, then since by definition of $n_2$ and $\epsilon$, this is $2e^{-\Theta\left(\frac{n_2 \rho^2 \nu}{\log^2(\nu^{-1})}\right)} \leq \frac{\min(\rho, \beta)}{10n_1}$. If $p < 3\nu$, this completes the proof. Otherwise, it is clear that the probability of $\text{Ev}_2(x)$ is maximized when $p(x) = 3\nu$, in which case since $\epsilon \leq \nu$, the above bound gives us $\Pr\left[\hat{p} \leq 2\nu\right] \leq \Pr\left[|p - \hat{p} \geq \epsilon|\right] \leq \frac{\rho}{10n_1}$.    $\square$

We next prove the following Lemma which will be used in our proofs.

**Lemma 18.** *Conditioned on the event that $\eta(x) \neq \eta(y)$ for all distinct $x, y \in X^{\geq \nu}$, the expected size of $X_{\overline{\eta}}^{\geq \nu}$ is at most $1 + \ln(\nu^{-1})$.*

*Proof of Lemma 18.* Let $x_i$ denote the value of $x \in X^{\geq \nu}$ with the $i$-the largest value for $p(x)$, where we assume that ties are broken arbitrarily. In order for $x_i$ to be in $X_{\overline{\eta}}^{\geq \nu}$, we need to have $\eta(y) > \eta(x_i)$ for all $y$ such that $p(y) \geq p(x_i)$, which in turn means that $\eta(x_i) < \eta(x_j)$ for all $j < i$. The probability that this happens is at most $1/i$ however since $\eta(.)$ is chosen uniformly at random and is independent across $x_i$. Formally,

$$\begin{aligned}
\Pr\left[x_i \in X_{\overline{\eta}}^{\geq \nu}\right] &= \Pr\left[\eta(y) > \eta(x) \; \forall y : p(y) \geq p(x_i)\right] &&\text{(Definition of } X_{\overline{\eta}}^{\geq \nu}) \\
&\leq \Pr\left[\eta(x_i) < \eta(x_j) \; \forall j < i\right] &&\text{(Since } p(x_j) \geq p(x_i) \text{ for } j \leq i) \\
&\leq 1/i &&\text{(Since } \eta(x) \text{ is chosen uniformly random).}
\end{aligned}$$

Since $X_{\overline{\eta}}^{\geq \nu} \subseteq X^{\geq \nu}$, summing over $i$ we obtain

$$\begin{aligned}
\mathbb{E}\left[|X_{\overline{\eta}}^{\geq \nu}|\right] &= \sum_{i=1}^{|X^{\geq \nu}|} \Pr\left[x_i \in X_{\overline{\eta}}^{\geq \nu}\right] &&\text{(Linearity of expectation)} \\
&\leq \sum_{i=1}^{|X^{\geq \nu}|} 1/i \leq 1 + \ln(|X^{\geq \nu}|).
\end{aligned}$$

Since $|X^{\geq \nu}| \leq \nu^{-1}$, the lemma follows.    $\square$

*Proof of Lemma 13.* For any distinct $x, y$, we have

$$\Pr\left[\eta(x) = \eta(y)\right] \leq 2^{-\Theta(\log(n_1 + \beta^{-1} + \rho^{-1}))} \leq \frac{\min(\rho, \beta)}{20n_1^2}.$$

Since $n_1 \geq \left|X^{\geq \nu}\right|$, It follows that with probability at least $1 - \min(\rho, \beta)/20$, the values of $\eta(.)$ are distinct across $X^{\geq \nu}$. Condition on this event. For any fixed value of $\eta$, since $\nu'$ is chosen uniformly at random from $[3/2\nu, 2\nu']$, we have $\Pr\left[\nu' \in U\right] \leq \frac{2\epsilon |X_{\overline{\eta}}^{\geq \nu}|}{\nu/2}$. Taking expectation over $\eta$, by Lemma 18 we have $\Pr\left[\nu' \in U\right] \leq \frac{2\rho\nu(\ln(\nu^{-1})+1)}{10\nu(\ln(\nu^{-1})+1)/2} \leq 2\rho/5$. Since We conditioned on an event with probability $1 - \min(\rho, \beta)/20$, we conclude that $\Pr\left[\text{Ev}_3\right] \geq (1 - \rho/20)(1 - 2\rho/5) \geq 1 - \rho/2$, finishing the proof.    $\square$

*Proof of Lemma 14.* For simplicity, we write $x^*$ instead of $x_{\eta,\nu'}$ for the rest of the proof. We first handle the case where $x^* \neq \texttt{Null}$. In this case, since $x^* \in X^{\geq \nu}$, we have $x^* \in S^{(1)}$ by $\texttt{Ev}_1$. We additionally claim that $x^* \in X_{\bar{\eta}}^{\geq \nu}$. If this is not the case, then there must be some $y \in X^{\geq \nu}$ such that $\eta(y) < \eta(x^*)$ and $p(y) \geq p(x^*)$. This implies however that $p(y) \geq \nu'$, which contradicts the definition of $x^*$ as the first elements (in order of $\eta(.)$) with probability at least $\nu'$. Since $x^* \in X_{\bar{\eta}}^{\geq \nu}$, we conclude that $p(x^*) \geq \nu' + \epsilon$ by $\texttt{Ev}_3$, which in turn implies $\hat{p}(x^*) \geq \nu'$ by $\texttt{Ev}_2$. It therefore suffices to show that $\hat{p}(x') \leq \nu'$ for all $x' \in S^{(1)}$ such that $\eta(x') < \eta(x^*)$.

By definition of $x^*$, we must have $p(x') \leq \nu'$, which further implies that $|p(x') - \hat{p}(x')| \leq \epsilon$ by $\texttt{Ev}_2$. We further assume that $p(x') \geq \nu$; if not, we have $\hat{p}(x') \leq \nu + \epsilon \leq \nu'$, finishing the proof. Therefore $x' \in X^{\geq \nu}$. Define $\widetilde{x}$ as the largest element before or equal to $x'$ in $X_{\bar{\eta}}^{\geq \nu}$; i.e., $\widetilde{x} = \arg\max_{x \in X_{\bar{\eta}}^{\geq \nu}: \eta(x) \leq \eta(x')} \eta(x)$. Note that $\widetilde{x} = x'$ if $x' \in X_{\bar{\eta}}^{\geq \nu}$.

We claim that $p(\widetilde{x}) \geq p(x')$. Assume this is not the case. Set $y$ to be the first element in $X^{\geq \nu}$ satisfying $p(y) > p(\widetilde{x})$. Formally, $y = \arg\min_{x \in X^{\geq \nu}: p(x) > p(\widetilde{x})} \eta(x)$. Since $p(x') > p(\widetilde{x})$ and $\eta(x') > \eta(\widetilde{x})$, we have $\eta(y) \leq \eta(x')$. We also have $\eta(y) > \eta(\widetilde{x})$; if not, then since $p(y) > p(\widetilde{x})$ we conclude that $\widetilde{x} \notin X_{\bar{\eta}}^{\geq \nu}$ which is a contradiction because $y \in X^{\geq \nu}$. It is easy to see however that $y \in X_{\bar{\eta}}^{\geq \nu}$, contradicting the definition of $\widetilde{x}$. Specifically, $p(y) > p(\widetilde{x})$ and $p(\widetilde{x}) \geq p(x)$ if $\eta(x) < \eta(y)$ by definition of $y$. It follows that the initial assumption was wrong and $p(\widetilde{x}) \geq p(x')$.

Given the above claim, we have $\eta(\widetilde{x}) \leq \eta(x') \leq \eta(x^*)$. Since $\widetilde{x} \in X^{\geq \nu}$, by definition of $x^*$, we have $p(\widetilde{x}) \leq \nu'$. Since $\widetilde{x} \in X_{\bar{\eta}}^{\geq \nu}$, by $\texttt{Ev}_3$ we conclude that $p(\widetilde{x}) \leq \nu' - \epsilon$, which further implies $p(x') \leq \nu' - \epsilon$. Therefore, by $\texttt{Ev}_2$ we have $\hat{p}(x') \leq \nu'$, finishing the proof.

If $x^* = \texttt{Null}$, then the same proof shows that $\hat{p}(x') \leq \nu'$ for all $x' \in S^{(1)}$; the only place where we used $x^*$ is for proving that $p(x') \leq \nu'$ and $p(\widetilde{x}) \leq \nu'$, both of which would hold trivially if $x^* = \texttt{Null}$. Therefore, output is $x^*$ in this case as well. □

## E  Omitted proofs

### E.1  Approximate median pair

In this section, we prove Proposition 3.

*Proof of Proposition 3.* We first invoke Lemma 2 to find a $(1/2 - \gamma/2)$-approximate median $\tau$ with the same sample complexity and parameters $\rho/4, \beta/4$. Let $\tau$ be the value obtained from the algorithm. Note that, if the output of the algorithm is correct, we have

$$\Pr[X \geq \tau], \Pr[X \leq \tau] > 1/2 - \gamma/2.$$

We then replicably estimate, with error $\gamma/8$, the value of $\Pr[X \geq \tau]$ and $\Pr[X > \tau]$. This can be done using, e.g., the SQ algorithm of [ILPS22]. Let $p_1, p_2$ denote the true probabilities and let $\hat{p}_1, \hat{p}_2$ denote our estimates. Assuming $\tau$ was chosen correctly, we have $p_1 \geq 1/2 - \gamma/2$ and $p_2 \leq 1/2 + \gamma/2$. This in turn implies that, if our estimates are correct, we must have $\hat{p}_1 \geq 1/2 - 3\gamma/4$ and $\hat{p}_2 \leq 1/2 + 3\gamma/4$. Now, if $\hat{p}_1 \leq 1/2 + 3\gamma/4$, then we output $(\tau, \kappa)$ for $\kappa = 0$. Otherwise, we set $\kappa$ such that $(1 - \kappa)\hat{p}_1 + \kappa\hat{p}_2 = 1/2 + 3\gamma/4$. It is clear that

$$(1 - \kappa)p_1 + \kappa p_2 = (1 - \kappa)\hat{p}_1 + \kappa\hat{p}_2 + (1 - \kappa)(p_1 - \hat{p}_1) + \kappa(p_2 - \hat{p}_2).$$

If our estimates are correct, it follows that $(1 - \kappa)p_1 + \kappa p_2$ is in the range $(1/2 - \gamma, 1/2 + \gamma)$, finishing the proof. □

### E.2  Replicable interior point hardness

In this section, we prove the hardness of the replicable interior point problem, as formalized in Proposition 5. We begin with some definitions. Recall that a randomized algorithm $A : \mathcal{X}^n \to \mathcal{Y}$ is called $(\varepsilon, \delta)$-DP (i.e., differentially private) if for any two multisets $S_1, S_2$ differing in exactly one coordinate, and any event $O \subseteq \mathcal{Y}$, we have

$$\Pr[A(S_1) \in O] \leq e^\varepsilon \Pr[A(S_2) \in O] + \delta,$$

where the probability is over the randomness of $A$. The following result from [BGH+23] shows that one can transform a replicable algorithm for a statistical problem to a differentially private algorithm for the same problem.

**Lemma 19** (Theorem 3.1 in [BGH+23]). *Let $\rho < 0.01$. If there is a $\rho$-replicable algorithm solving a statistical problem with failure probability $\beta$ and sample complexity $n$, then there is an $(\varepsilon, \delta)$-DP algorithm solving the same statistical problem with failure probability at most $O(\beta \log 1/\beta)$ and sample complexity $n \cdot O(1/\varepsilon \log 1/\delta \log 1/\beta + \log^2 1/\beta)$.*

Existing works on differential privacy show that, for a closely related problem, any differentially private algorithm requires at least $\Omega(\log^* |\mathcal{X}|)$ samples. We will refer to this problem as the *non-statistical interior point problem* and define it as follows. Given a set of samples $S \subseteq \mathcal{X}$, define its interior as the set $\texttt{Interior}(S) = [\min(S), \max(S)]$. In the *non-statistical* interior point problem, we are given a set $S$ and the goal is to output some point $x$ in its interior. To avoid confusion, throughout the section we will refer to our main problem, in which the goal is to find a point in the interior of a distribution using samples, as the *statistical* interior point problem.

We say an algorithm $A$ has sample complexity $n$ for the non-statistical interior point problem with error probability $\beta$ if, for any $S \subseteq \mathcal{X}$ of size $n$, we have $\Pr[\mathcal{A}(S) \in \texttt{Interior}(S)] \geq 1 - \beta$, where the probability is over the randomness of the algorithm. The following result by Bun et al. [BNSV15] provides a lower bound for the sample complexity of differentially private algorithms on this problem.

**Lemma 20** ([BNSV15]). *Fix $\varepsilon \in (0, 1/4)$. Assume that $\delta(n) \leq 1/(50n^2)$. For any $n$, solving the non-statistical interior point problem with $(\varepsilon, \delta(n))$ differential privacy requires sample complexity $n \geq \Omega(\log^* |\mathcal{X}|)$.*

We next show how to transform an algorithm for statistical interior point to an algorithm for non-statistical interior point with a similar guarantee on differential privacy.

**Lemma 21.** *Let $\varepsilon < 1/2$. If there is an $(\varepsilon, \delta)$-DP algorithm for the statistical interior point problem with error probability $\beta$, there is an $(\varepsilon', \delta')$-DP algorithm for non-statistical interior point with sample complexity and error probability $\beta$ where $\varepsilon' = O(\varepsilon \log 1/\delta)$ and $\delta' = O(\delta)$.*

*Proof.* Let $A$ be the mentioned statistical algorithm. Consider the following algorithm $A'$. Given the set $X$ of size $n$, we let $S$ be a set of $n$ i.i.d. samples from the distribution $\hat{D} = X$, and output $A(S)$. We show that $A'$ satisfies the required guarantees.

**Correctness analysis.** With probability $1 - \beta$, the algorithm outputs a value in the interior of $D$, i.e., the range $[\min \hat{D}, \max \hat{D}]$, which is the same as $[\min(X), \max(X)]$.

**Privacy analysis.** Let $X_1, X_2 \in \mathcal{X}^n$ be two vectors that differ in one entry. Let $\mathcal{Y}$ denote the output space of $A$. We need to show that for any $O \subseteq \mathcal{Y}$, we have

$$\Pr[A'(X_1) \in O] \leq e^{\varepsilon'} \Pr[A'(X_2) \in O] + \delta',$$

where the probability is over the randomness of $A'$. Let $\hat{D}_1$ and $\hat{D}_2$ be the corresponding empirical distributions for $X_1$ and $X_2$. By definition of $A'$, the above inequality is equivalent to

$$\Pr_{S \sim \hat{D}_1}[A(S) \in O] \leq e^{\varepsilon'} \Pr_{S \sim \hat{D}_2}[A(S) \in O] + \delta', \tag{3}$$

where the randomness is now over both the sampling of $S$ and the internal randomness of $A$.

We may assume that the set $S$ is sampled as follows. We first sample a multiset of indices $U \subseteq [n]$ with size $n$ by uniformly sampling $n$ values i.i.d. from $[n]$. Note that this means an index may appear multiple times in $U$. We then obtain the $i$-th entry of $S$ by taking the $j$-th entry of either $X_1$ or $X_2$, depending on the input, where $j$ is the $i$-th entry in $U$. It is clear that this does not change the sampling procedure. Using this view, however, we can now couple the two probabilities corresponding to $X_1$ and $X_2$ in order to compare them. Formally, let $A(U; X)$ denote the value of $A(S)$ when $S$ is obtained from the above procedure. We need to compare the values $\Pr[A(U; X)]$ for $X = X_1$ and $X = X_2$, where the randomness is over the draw of $U$ and the internal randomness of $A$.

Assume without loss of generality that $X_1, X_2$ differ on their first coordinate. Let $I$ denote the number of times the value 1 appears in $U$. Set $t = 10 + \ln 1/\delta$. By iterated expectation,

$$\Pr\left[A(U; X) \in O\right] = \sum_{i=1}^{n} \Pr\left[I = i\right]\Pr\left[A(U; X) \in O \mid I = i\right]$$

$$\leq \sum_{i=1}^{t} \Pr\left[I = i\right]\Pr\left[A(U; X) \in O \mid I = i\right] + \Pr\left[I \geq 1 + t\right].$$

Since $\mathbb{E}\left[I\right] = 1$, Chernoff's inequality gives

$$\Pr\left[I \geq 1 + t\right] \leq \frac{e^t}{(1 + t)^{1+t}} \qquad \textit{(Chernoff's bound)}$$

$$\leq e^{-t} \qquad \textit{(Since } t \geq 10\textit{)}$$

$$\leq \delta \qquad \textit{(Since } t \geq \ln 1/\delta\textit{)}.$$

It follows that

$$\Pr\left[A(U; X_1) \in O\right] \leq \sum_{i=1}^{t} \Pr\left[I = i\right]\Pr\left[A(U; X_1) \in O \mid I = i\right] + \delta. \qquad (4)$$

We next claim that

$$\Pr\left[A(U; X_1) \in O \mid I = i\right] \leq e^{i\varepsilon}\Pr\left[A(U; X_2) \in O \mid I = i\right] + \delta_i, \qquad (5)$$

where $\delta_i = \delta\frac{e^{i\varepsilon} - 1}{e^\varepsilon - 1}$. Formally, let $\mathcal{U}_i \subseteq [n]^n$ denote the set of all ordered vectors of length $n$ from $[n]$ that have exactly $i$ entries of 1. By total probability,

$$\Pr\left[A(U; X) \in O \mid I = i\right] = \sum_{u \in \mathcal{U}_i} \Pr\left[U = u \mid I = i\right]\Pr\left[A(u; X) \in O\right],$$

where the randomness in the probability in the right hand side is now only the randomness of the algorithm $A$. We now note that for all $u \in \mathcal{U}_i$,

$$\Pr\left[A(u; X_1) \in O\right] \leq e^{i\varepsilon}\Pr\left[A(u; X_2) \in O\right] + \frac{e^{i\varepsilon} - 1}{e^\varepsilon - 1}\delta.$$

This holds because $A$ is $(\varepsilon, \delta)$-DP and the set $S$ given as input to $A$ in $A(u; X_1)$ and $A(u; X_2)$ differ in exactly $i$ coordinates. Summing over all $u$, we obtain Equation (5).

Plugging this in Equation (4), we obtain

$$\Pr\left[A(U; X_1) \in O\right] - \delta$$

$$\leq \sum_{i=1}^{t} \Pr\left[I = i\right]\left(e^{i\varepsilon}\Pr\left[A(U; X_2) \in O \mid I = i\right] + \delta_i\right)$$

$$\leq \sum_{i=1}^{t} e^{i\varepsilon}\Pr\left[I = i\right]\Pr\left[A(U; X_2) \in O \mid I = i\right] + \sum_{i=1}^{t} \Pr\left[I = i\right]\delta_i$$

$$\leq \left(e^{\varepsilon t}\sum_{i=1}^{t} \Pr\left[I = i\right]\Pr\left[A(U; X_2) \in O \mid I = i\right]\right) + \left(\sum_{i=0}^{n} \Pr\left[I = i\right]\delta_i\right) - \delta_0,$$

where in the last inequality we have used the fact that $i \leq t$ to bound the first expression and the non-negativity of $\delta_i$ for $i > 1$ to bound the second expression. By total probability, the first expression is at most $e^{\varepsilon t}\Pr\left[A(U; X_2) \in O\right]$. As for the second expression, it is equal to $\mathbb{E}\left[\delta_I\right] = \frac{\delta}{e^\varepsilon - 1}\mathbb{E}\left[e^{\varepsilon I} - 1\right]$. Observe however that $I$ follows a binomial distribution with parameters $n$ and $1/n$. Therefore,

$$\mathbb{E}\left[e^{\varepsilon I}\right] = \left(1 - \frac{1}{n} + \frac{e^\varepsilon}{n}\right)^n$$

$$\leq \left(1 + \frac{2\varepsilon}{n}\right)^n \qquad \textit{(Since } e^x - 1 \leq 2x \text{ for } x < 1/2 \text{ and } \varepsilon < 1/2\textit{)}$$

$$\leq e^{2\varepsilon}.$$

It follows that

$$\begin{aligned}
\mathbb{E}\left[\delta_I\right] &\leq \frac{\delta}{e^\varepsilon - 1}\left(e^{2\varepsilon} - 1\right) \\
&\leq \delta(e^\varepsilon + 1) \\
&\leq (e^{1/2} + 1)\delta \qquad\qquad\qquad \textit{(Since } \varepsilon < 1/2\textit{)} \\
&\leq 3\delta.
\end{aligned}$$

Finally, we have $\delta_0 = 0$. Plugging this back in we obtain Equation (3) with $\delta' = 4\delta$ and $\varepsilon' = (\ln(1/\delta) + 10)\varepsilon$, finishing the proof. $\qquad\square$

We now prove Proposition 5.

*Proof of Proposition 5.* Assume that an algorithm with these properties exists. Let $n$ denote its sample complexity. For any $\varepsilon, \beta$, by Lemma 19 we have an $(\varepsilon, \delta)$-DP algorithm for statistical interior point with failure probability $O(\beta \log 1/\beta)$ and sample complexity

$$n \cdot O\left(\frac{1}{\varepsilon}\log\frac{1}{\delta}\log\frac{1}{\beta} + \log^2\frac{1}{\beta}\right).$$

By Lemma 21, this means that there is an $(\varepsilon', \delta')$-DP algorithm for non-statistical interior point with failure probability $\beta'$ and the same sample complexity where

$$\varepsilon' = O\left(\varepsilon\log\frac{1}{\delta}\right), \quad \delta' = O(\delta), \quad \beta' = O\left(\beta\log\frac{1}{\beta}\right).$$

Set $\beta$ such that $\beta' = 0.05$. Note that this can be achieved for some constant value $\beta > 0$. We can obtain any desired value for $\varepsilon', \delta'$ by setting $\delta' = \Theta(\delta)$ and $\varepsilon = \Theta(\varepsilon'/\log 1/\delta')$. It follows that, for any $\varepsilon', \delta'$, there exists an $(\varepsilon', \delta')$-DP algorithm with failure probability $0.05$ and sample complexity $N = O\left(n(1 + 1/\varepsilon'\log^2 1/\delta')\right)$ for some constant $c$. Set $\varepsilon' = 0.05$. If $\delta' = \frac{1}{c'n^3}$, then this gives us the sample complexity $N = cn(1 + \log^2(c'n^3))$. We claim that if we choose $c'$ to be large enough, we can ensure that $\delta' \leq 1/50N^2$. By definition of $\delta'$, this means we need to show that

$$c'n^3 \geq 50c^2n^2(1 + \log^2(c'n^3))^2.$$

Given the inequalities $(x + y + z)^2 \leq 3(x^2 + y^2 + z^2)$ and $(x + y)^2 \leq 2(x^2 + y^2)$ we have

$$\begin{aligned}
\frac{1}{3}(1 + \log^2(c'n^3))^2 &= \frac{1}{3}(1 + (\log(c') + \log(n^3))^2)^2 \\
&\leq \frac{1}{3}(1 + 2\log^2(c') + 2\log^2(n^3))^2 \\
&\leq 1 + 4\log^4(c') + 9\log^4(n)
\end{aligned}$$

It is sufficient to show that

$$c'n \geq 150c^2\left(1 + 4\log^4 c' + 9\log^4 n\right).$$

Setting $c'$ to be large enough in terms of $c$, the above inequality holds as each term on the right hand side is asymptotically smaller than the left hand side.

Therefore, we have obtained an $(\varepsilon', \delta')$-DP algorithm with sample complexity $N$ and failure probability $\beta'$ where $\delta' \leq 1/50N^2$, and $\varepsilon', \beta' \leq 0.05$. Invoking Lemma 20, we conclude that $N \geq \Omega(\log^*|\mathcal{X}|)$. Since $\delta' = \Theta(1/n^3)$, this implies that $n\log^2 n \geq \Omega(\log^*|\mathcal{X}|)$, which in turn implies $n \geq \tilde{\Omega}(\log^*|\mathcal{X}|)$ as claimed. $\qquad\square$

