# OpenReview forum: "Replicable Online pricing"
_NeurIPS.cc/2025/Conference — NeurIPS 2025 poster_

### Official Review · Reviewer_Vyhk · 2025-06-30

**Clarity:** 3
**Significance:** 3
**Originality:** 3
**Rating:** 5
**Confidence:** 1

**Summary:**

The authors study replicable online pricing, a variant of prophet inequalities in which the algorithm does not have direct access to the underlying distributions but can sample from them, and it is required that the acceptance threshold must be chosen so that the output is stable under resampling ($\rho$-replicable).
For the case where the distributions are finitely supported, the authors design a pricing rule that is $(1/2-\varepsilon)$-competitive using $\textrm{poly}(\log^*(|X|),1/\rho,1/\beta,1/\varepsilon)$ samples, where $X$ is the finite support of the value distributions.

The authors also show via a reduction from replicable interior-point estimation that the $\log^{*}(|X|)$ dependence is unavoidable.
Beyond prophet inequalities, the paper extends the techniques to a delegation model, obtaining the same sample complexity, and introduces a new replicable heavy-hitter algorithm whose sample complexity is nearly linear in $1/\nu$, where $\nu$ is the heavy-hitting parameter, improving on prior cubic bounds in $1/\nu$.

**Questions:**

Your sample-complexity bound is polynomial in $\rho^{-1}, \beta^{-1}, \varepsilon^{-1}$, but you don’t give any matching lower bounds in these parameters. Do you have evidence that this polynomial blow-up is supposed to be there, or do you suspect those exponents could be shaved, maybe even down to log-level?

Typos: line 217 "we say that \tau is \alpha-competetive if..."

**Ethical Concerns:**

["NO or VERY MINOR ethics concerns only"]

**Final Justification:**

I'm not familiar with the line of work this paper is about, but I enjoyed reading it and to me it's a valuable contribution. Hence, even if with low confidence, I recommend the paper for acceptance.

**Limitations:**

yes

**Quality:**

3

**Strengths And Weaknesses:**

Strenghts:
1) Near-optimal sample bound for replicable prophet inequalities and delegation.
2) Improved algorithms for the replicable heavy-hitter problem.

Weaknesses:
1) as noted in footnote 2, the proposed algorithm implicitly relies on correlated sampling and hence is not computationally efficient.

---

> ### Author Rebuttal · Authors · 2025-07-30
>
> Thank you for your positive review!
> Below we address your individual comments and questions.
>
> > … don’t give any matching lower bounds in these parameters … evidence that this polynomial blow-up is supposed to be there
>
> Thank you for the question. In obtaining our result, we primarily had in mind the constant-parameter regime where $\rho$, $\beta$, and $\epsilon$ are set to small constants (e.g., 0.01). That said, we agree that understanding the precise dependence on these parameters is an interesting direction, both for our problem and for replicable median estimation more broadly. We will add this to the discussion of future work.
> Regarding the dependence on $\beta$, we recently realized that it can be improved to polylogarithmic. We will revise our paper to reflect this; the improvement follows by tracking parameter dependence in the proof sketch of Lemma 2 in [BNH+23]. For the other two parameters, we suspect that a polynomial dependence is necessary, though we do not currently have a proof. In the case of $\rho$, most prior work on replicability incurs a $\rho^{-2}$ dependence, and for some problems (e.g., mean estimation), this is known to be necessary (see Theorem 7.1 in [ILPS22]). As for $\epsilon$, the dependence is likely to be polynomial when using a median-based approach such as ours, given the known sample complexity of estimating the median even without replicability. Moreover, our lower bound in Theorem 3 relates online pricing to the interior point problem, which is closely tied to median estimation, and suggests that any algorithm for our setting may inherently need to perform some form of median estimation. While this connection is suggestive, we do not have a formal proof establishing it.

---

> > ### Comment · Reviewer_Vyhk · 2025-08-09
> >
> > Thank you for the answers. I confirm my score and I'll take your rebuttal into account while discussing with the other reviewers.

---

### Official Review · Reviewer_mjxJ · 2025-07-01

**Clarity:** 3
**Significance:** 3
**Originality:** 3
**Rating:** 5
**Confidence:** 2

**Summary:**

The paper analyzes the problem of Online Pricing where the goal is to set a threshold $\tau$ such that the reward obtained by accepting whatever first value received that exceeds $\tau$ is at least $\alpha$-approximation of the maximum of the sequence with probability at east $1-\beta$. In particular the paper focuses on replicable algorithms to design this threshold, as in algorithm where the output is determined by its input more than the internal randomization. Furthermore, they analyse the delegation problem, where a principal has to set a threshold such that the value reward chosen from him by an agent among a set of possible values exceeding the threshold is maximized. The results provided characterize the sample complexity needed in both problems to reach a $1/2 - \epsilon$ competitive ratio. Furthermore they show that in the case of the Online Pricing problem the dependency on the support of the distribution in the sample complexity is unavoidable. Lastly,  they provide results for the same problem, in the case in which the samples are chosen in an i.i.d. fashion and design an algorithm that recognize the heavy hitters of the distribution and provide, in this case as well bounds on its sample complexity

**Questions:**

1. Does an analogous result of Theorem 3 holds even for the delegation problem? Or more in general are you aware if the sample complexity expressed in Theorem 2 is tight?

2. Similarly for Theorem 5, is the result provided tight?

**Ethical Concerns:**

["NO or VERY MINOR ethics concerns only"]

**Final Justification:**

I think this paper is valid, hence I keep my positive score unchanged.

**Limitations:**

yes

**Paper Formatting Concerns:**

I have not noticed any major formatting issues. I noticed the following typos:

Line 217 - There is a missing parenthesis in the left hand expectation;

**Quality:**

3

**Strengths And Weaknesses:**

The paper analyzes a well know theoretical problem (prophet inequality) which is well-established in the machine learning community, therefore of general interest. The results provided are sound, bringing new insights about the topic. Moreover, the focus on replicability adds originality to the results. Lastly, all the assumptions required are clearly stated.

On another note, I think the structure of the paper could be improved: having the statements of the theorems before the setting and the notation is introduced, makes the paper sometimes hard to read.

---

> ### Author Rebuttal · Authors · 2025-07-30
>
> Thank you for your positive review!
> Below we address your individual comments and questions.
>
>
>
> > I think the structure of the paper could be improved …
>
> Thank you for the helpful feedback. We will make sure to incorporate your suggestion for the final version of our paper.
>
> > Does an analogous result of Theorem 3 holds even for the delegation problem?
>
>
> Thank you for the question. We initially focused on the prophet inequalities problem in our lower bound, as it is the primary focus of the paper. However, you are correct that the argument can be extended to the delegation problem as well, and we will add this to the final version. The extension follows by establishing an analogue of Lemma 8 for the delegation setting, using a similar proof.
>
>
> > ... Theorem 5, is the result provided tight?
>
>
> Examining the dependence on individual parameters, the $\nu^{-1}$ term is necessary: even without replicability, identifying a single $\nu$-heavy hitter requires at least $\Omega(\nu^{-1})$ samples. As for the $\rho^{-2}$ dependence, while we do not know whether it is necessary for this specific problem, all known algorithms exhibit this dependence in the worst-case. More broadly, a $\rho^{-2}$ term commonly arises in the replicability literature and for certain problems such as mean estimation, it is necessary (see Theorem 7.1 of [ILPS22]).

---

> > ### Comment · Reviewer_mjxJ · 2025-08-04
> >
> > I thank the authors for their careful replies to the points I have raised.
> >
> > I truly recommend implementing the changes you propose, as it would benefit both the clarity and completeness of the paper. Overall, I remain of the idea that this is a valid work and I will keep my score unchanged.

---

### Official Review · Reviewer_ZNhf · 2025-07-01

**Clarity:** 3
**Significance:** 2
**Originality:** 2
**Rating:** 3
**Confidence:** 3

**Summary:**

The authors explore, for the first time, the concept of replicability for two important online decision-making problems, which sit at the intersection of computer science and economics; the classic prophet inequality and the delegation problem. Replicability is a well-established algorithmic notion in statistical learning, which intuitively states that the algorithm’s output is determined primarily by the input distribution rather than specific samples and/or other sources of randomness. It has been used so far to produce replicable algorithms in important problems of statistical/machine learning.

The authors focus on threshold-based algorithms and study the sample complexity (since, as is often assumed, the distributions are not known but we have sample access to them) and competitive guarantees of algorithms they design or inherit from previous works. By designing efficient replicable algorithms and establishing lower and upper bounds for the sample complexity (with some factors in the expression coming from the definition of the replicable online pricing problem) for the two settings, they showcase the trade-offs between replicability, competitiveness, and sample efficiency. More specifically, they give a replicable algorithm for the prophet inequality with an almost tight competitive ratio with high probability that requires relatively few samples from each distribution; they subsequently extend this result to the delegation problem. They then show that the sample complexity cannot be independent of the support size of the product distribution. To prove this, they develop a novel and improved replicable algorithm for outputting a heavy hitter for an input distribution (this is a slightly less general version of a well-known problem studied in related works), which may be of independent interest. They also provide a result that, in the iid case and for specific regimes, gives a tight factor for the dependence of the sample complexity on the competitive ratio.

**Questions:**

I got a bit confused from the abstract, which one is the matching lower bound? I understood that the authors show the bound has to depend on $\log^*\mathcal{\mid X \mid}$, but they don't match it up to the factor of the upper bound (?) Please clarify this, and also what happens with the other factors, so that you can have a tight bound.

I also got a bit confused in theorem 4, why is $\alpha<=¼$? Does this mean that this the regime for which the dependence on $\alpha$ is tight, but is $\alpha>¼$ we don't know?

It was a bit hard to follow all steps in the appendix for the proof of Theorem 3 (including the result for the replicable heavy hitter); is it the case that you cannot use as black box the previous algorithms  for the heavy hitters problem to show the dependence on $\log^*\mathcal{\mid X \mid}$, i.e., the new algorithm with the improved factor is necessary?

Minor notes:

In Lemma 2, maybe at least cite the exact lemma from the other paper, or give the expression with the various polynomial factors on $\log^*\mathcal{\mid X \mid}, 1/\rho, 1/\beta,  1/\epsilon$, to make it self-contained and know what these are. And once given, you can use the $poly()$ expression for ease of exposition.

If possible, expand a bit more on the significance of Theorem 4: Why do we need to know the exact dependence on $\alpha$?

**Ethical Concerns:**

["NO or VERY MINOR ethics concerns only"]

**Final Justification:**

The authors have mostly addressed the questions I asked in the rebuttal. I think it's an interesting topic and the authors provide a satisfactory answer to the main question they ask. However, I'm unsure about the novelty in parts of the paper, which makes my score borderline. I also think that the paper should be revised/polished, so that the new contributions and techniques and the relation to previous results can become more clear.

**Limitations:**

yes

**Paper Formatting Concerns:**

-

**Quality:**

2

**Strengths And Weaknesses:**

Strengths:

Studying replicability in online decision-making is an interesting research direction, especially in important practical scenarios that lack replicability guarantees.

It's good that the algorithms for the two problems can give competitive ratios that are almost tight, while not requiring too many samples from each distribution; all factors in the sample complexity expression are essentially constant (since the iterated logarithm is always very small, even for relatively large support sizes).

The result that shows that the dependence of the sample complexity on $\log^*\mathcal{\mid X \mid}$ is necessary is interesting and the proof involves going through various non-trivial steps, including the new algorithm for the heavy-hitter problem.

Weaknesses:

I think what the authors do not mention explicitly is that from the work of Samuel-Cahn we know that setting the median as threshold also gives the optimal $½$-approximation for prophet inequalities (apart from setting half of the expectation of the max as threshold). Thus, the algorithm used is known, and since the replicable algorithm for approximate median estimation is from previous work, Theorem 1 becomes more of an observation, because it just combines the previous result with the known algorithm to get a replicable algorithm for the prophet inequality. For Theorem 2, we know from the work of Kleinberg and Kleinberg that the delegation problem is closely connected to the classic prophet inequality, so it is expected that the guarantees carry over to this setting and its slight variants.

Apart from Theorem 3, which I mentioned in the strengths, it doesn't seem that the rest of the results can account for significant contributions either. Theorem 4 goes into the case of iid distributions to show a tight linear dependence on  $\alpha$ (i.e., the competitive ratio), which is nice to know, but it's also just one of the constant factors, so the gain is to know that the tight bound will depend linearly on this specific constant. However, it’s just a small factor in the number of samples that we need. Similarly, concerning theorem 5, it's interesting that en route to prove theorem 3 the authors get an improved sample complexity for this version of the heavy hitter problem, but, as they say, they improve a bit on one factor, and some of the improvement comes already  from the concentration bound that doesn't apply to the more general setting the previous works studied. Also, they acknowledge that their two-stage approach has been previously explored, so I'm not sure if there's significant gain in terms of novel techniques that can be applied to this, or similar, problems.

Still, in practical scenarios, the single-sample algorithm of [RWW19] is very simple and optimal. I get the argument that it doesn't work with the definition of replicability that the authors follow, but it's interpretable, very easy to implement, recovers the optimal guarantee with full knowledge of the distributions, and requires just one sample from each distribution. Given that, maybe there are other online decision-making problems for which a replicable solution makes for a stronger case.

---

> ### Author Rebuttal · Authors · 2025-07-30
>
> Thank you for your careful reading of our paper and feedback.
> Below we address your individual comments and questions.
>
>
> > “…matching lower bound…”
>
> Here by matching lower bound, we intended to imply a polylog-star dependence on $|X|$ as in the place we claim an upper bound of $\text{poly}(\log^* |X|)$ and a lower bound of $\log^* |X|$. We certainly understand your concern and will revise the terminology to avoid any possible confusion. Regarding the other parameters, it is standard to take these parameters to be small constants.
>
> > “…why is $¼$…”
>
> Note that the theorem also holds for $\alpha \in (1/4, 1/2 - \epsilon)$ by Theorem 1, which considers a more general (not necessarily i.i.d.) setting. We assumed $\alpha \le 1/4$ in Theorem 3 to simplify the proof. One cannot obtain $\alpha$ arbitrarily close to $1$ because of existing hardness results for the non-replicable version of the problem. Our main focus is small values of $\alpha$ as we want to show the linear dependence on $\alpha$ in Theorem 3 is tight.
>
> > “…use previous algorithms…”
>
> Our improved bounds are necessary for the range we consider. Specifically, if the replicable heavy hitter algorithm has sample complexity $\nu^{-3}$ instead of $\nu^{-1}$, then in Theorem 3 we can only handle $\alpha \ge \Theta((\log^* |X|)^{-0.33})$ instead of $\Theta((\log^* |X|)^{-0.99})$.  The improved bound we prove allows us to handle a much larger range.  We will further emphasize this point in the paper.
>
> > “…expression with the various polynomial factors…”
>
> Thank you for the suggestion. Following the argument in Footnote 18 of the arXiv version of [BGH+23] and keeping track of the relevant parameters, we obtain the bound $\tilde{O}(\\epsilon^{-4} \rho^{-2} \log^2(1/\beta) (\log^*|X|)^3)$.  We will further revise our paper to reflect this.
>
>
> > “…significance of Theorem 4…”
>
> Here we intend to show that the linear dependence on $\alpha$ in Theorem 2 is tight. Additionally, the theorem is conceptually interesting as it shows that, if we are willing to settle for smaller values of $\alpha$, we can obtain replicable prices using fewer samples.
>
> > “…work of Samuel-Cahn…”
>
> We assumed that the work of Samuel-Cahn is folklore in the community. We will certainly clarify this in our paper to avoid any confusion for readers outside the field.
>
> If you feel that our response has adequately addressed your major concerns, we would appreciate it if you possibly adjust your score accordingly.

---

> > ### Author Response · Authors · 2025-08-03
> >
> > Dear Reviewer,
> >
> > We would like to kindly check whether you have any remaining questions or concerns. We would be happy to provide further clarification. If not, we would be grateful if you could consider updating your score in light of the rebuttal.

---

> > ### Comment · Reviewer_ZNhf · 2025-08-04
> >
> > Thank you for your replies to my questions. I remain borderline about the paper, since I still see the result on the replicable algorithm for prophet inequalities quite straightforward, given what we know from previous works. Also, given the many (positive) results on sample-based prophet inequalities, I'm not sure why we would prefer an algorithm with this sample complexity in practice.
> > On another note, I see that two of the reviewers mention the novel algorithm for a variant of the heavy hitter problem in the strenghts, and as an important contribution of its own. I didn't see it as such in my review, however, I'm not very familiar with this problem and the past contributions, so I may be missing something here.

---

> > > ### Author Response · Authors · 2025-08-05
> > >
> > > Thank you for your response.  Below, we address two specific points regarding the heavy hitter result and the sample complexity of our algorithms.
> > >
> > > **On the heavy hitter result**:
> > > The heavy hitter problem is a fundamental and widely studied task, with classic works such as [1, 2] receiving thousands of citations. Given its importance, we believe it is valuable to develop a toolbox of efficient replicable algorithms for natural and motivated variants. Our result contributes to this direction.
> > >
> > > **On sample complexity**:
> > > Increased sample complexity is inevitable in replicable algorithm design. For example, even basic tasks like replicable median estimation require more samples than their non-replicable counterparts. Our results incur only a modest increase. In our view, this is justified because replicability is an important and well-motivated constraint, and designing algorithms that satisfy it is a meaningful goal. Naturally, as with any constraint, enforcing replicability may increase sample complexity. We believe this trade-off is worth studying, and that a large part of the machine learning community, and even scientific community shares this view, as evidenced by the numerous papers and workshops devoted to the topic.
> > >
> > >
> > > [1] Moses Charikar, Kevin Chen, and Martin Farach-Colton. Finding frequent items in data streams. In, ICALP 2002.
> > >
> > > [2] Noga Alon, Yossi Matias, and Mario Szegedy. The space complexity of approximating the frequency moments. In, STOC 1996.

---

> > > > ### Comment · Reviewer_ZNhf · 2025-08-07
> > > >
> > > > Thank you for the additional clarifications. I do believe that the question that the paper asks in novel and interesting and that it offers new insights into a well-established decision-making problem. On the other hand, I feel that the paper could benefit from some (major) rewriting so that 1) its main body becomes a bit easier to read and 2) it becomes clearer what is known already and directly adapted from previous works (in terms of results and techniques) and the main contributions of the paper are better showcased (e.g., how the new techniques and adaptations in the improved result for the heavy hitter problem can be of broader interest, which of the techniques used for the prophet inequality problem can serve as a toolbox for developing replicable algorithms for other interesting problems). For the current submission, I tend to keep my original score.

---

### Official Review · Reviewer_5wxJ · 2025-07-03

**Clarity:** 3
**Significance:** 4
**Originality:** 3
**Rating:** 4
**Confidence:** 2

**Summary:**

This paper studies replicable algorithms for the Prophet inequality and delegation problems. It focuses on a sample-based version of the Prophet inequality, aiming to characterize the sample complexity required for replicable algorithms to find a threshold (or price) that ensures competitiveness. The authors propose a median-based approach that is both replicable and near-optimal. They then extend this approach to the delegation problem and further establish that the dependence of the sample complexity on the size of the distribution’s support is tight.

**Questions:**

- Related to the weakness mentioned above: could the authors provide a separate section or dedicated paragraphs summarizing existing algorithms and results, including (but not limited to) the state of the art in sample-based Prophet inequalities, mean and median estimation algorithms, and the heavy hitter problem?

- In Section 3, it is stated that using mean estimation introduces additive error, which is undesirable. Could the authors elaborate on this point or provide supporting references?

- In many practical scenarios, samples are expensive. Given the increased sample complexity required for replicability, especially in pricing problems, how do the authors justify that replicability outweighs this cost?

**Ethical Concerns:**

["NO or VERY MINOR ethics concerns only"]

**Final Justification:**

The authors have partially addressed my concerns regarding the presentation and the sample complexity. I will keep my score, as it already reflects my evaluation of the paper: it studies a classic problem using a new (though not original) framework.

**Quality:**

4

**Strengths And Weaknesses:**

Strength
- Designing replicable algorithms is of significant importance for decision-making problems. This paper provides interesting results for the well-known Prophet inequality and demonstrates a fundamental trade-off between sample complexity, competitiveness, and replicability.

- The theoretical contributions are strong, with both upper and lower bounds rigorously established.

Weakness

- The discussion of related work and prior results is not clearly separated from the authors’ own contributions. References to existing algorithms and techniques are often interwoven with the presentation of new results, which makes it challenging to understand what is new. It can enhance the clarity and readability of the paper by introducing a separate section or paragraph that clearly summarizes relevant prior work (The related work in Appendix does not include sufficient discussions about the algorithmic ideas and techniques).

---

> ### Author Rebuttal · Authors · 2025-07-30
>
> Thank you for your careful reading of our paper and feedback.
> Below we address your individual comments and questions.
>
> > “…could the authors provide a separate section or dedicated paragraphs summarizing existing algorithms and results…”
>
> Our paper already discusses relevant prior work in Appendix A, including known
> results on replicable algorithms for mean and median estimation, as well as the
> heavy hitter problem. That said, we understand the reviewer’s concern about
> clarity and agree that separating background from our contributions would make
> the paper easier to follow.
>
> To address this, we will add the following paragraphs to the paper. If there are any other important references you believe should be discussed, please let us know and we would be happy to include them.
>
>
> **Sample-based prophet inequalities.** The sample-based study of prophet inequalities—where the algorithm has access only to samples from the underlying distributions—has become an active and evolving area of research. This direction was initiated by Azar, Kleinberg, and Weinberg [1], who drew connections between this setting and the classical secretary problem. Specifically, they provide a general reduction, showing that any algorithm in the order-oblivious secretary setting implies an algorithm in the single-sample prophet inequalities setting. For the case of single-choice prophet inequalities problem, Rubinstein, Wang, and Weinberg [2] showed that even a single sample per distribution suffices to achieve the optimal 1/2-approximation. In the i.i.d. variant, they also proved that a constant number of samples is enough, with later work by Correa et al. [3] refining the required sample complexity. When the arrival order is random, as in the prophet secretary model, Correa et al. [4] established that a single sample still enables a 0.635-approximation. This sampling approach has since been extended to more structured selection problems, such as those involving matroids and matchings [5, 26]. More recently, Cristi and Ziliotto [7] developed a unified argument showing that constant sample access suffices to achieve approximate optimality across both prophet-secretary and free-order settings, despite the exact optimal approximation ratios for these variants remaining open.
>
> **Replicable mean and median estimation.**  Replicable algorithms for mean and
> median estimation were first developed by [8], who introduced the notion of replicability and  provided replicable algorithms for several important problems. Their replicable median algorithm has sample complexity exponential in $\log^* |X|$.
> [9] later improved on this algorithm by drawing connections to differential privacy, obtaining an algorithm with sample complexity that is polynomial in $\log^* |X|$, albeit without computational efficiency.
>
> **Replicable heavy hitters.**  Prior work has also studied replicable
> algorithms for heavy hitter identification. [8] and [10] provide
> algorithms that return a list of candidate heavy hitters, with sample
> complexity scaling cubically in $1/\nu$.
> More recently, [11] improved the dependence on the failure and replicability parameters for expected sample complexity,
> but still maintained a $\nu^{-3}$ dependence.
> In contrast, we consider a more specialized variant where only a single heavy hitter needs to be output (assuming one exists), and give a new algorithm with nearly linear $\nu^{-1}$
> dependence.
>
>
>
>
>
> > ... introduces additive error, which is undesirable ... elaborate on this point or provide supporting references
>
> We focus on multiplicative error, as it is the standard metric for evaluating algorithmic performance in the context of prophet inequalities (e.g., see [1–7, 12–15]). Multiplicative guarantees are preferred over additive ones because they remain meaningful even when the offline optimum is small. For example, an algorithm that achieves expected value within $\epsilon$ of the offline optimum $\textnormal{OPT}$ (i.e., additive error) offers no useful guarantee when $\textnormal{OPT} < \epsilon$. In contrast, a multiplicative guarantee ensures that the algorithm's performance is always an approximation of the optimum, regardless of its magnitude.
>
>
>
>
> > …the increased sample complexity required for replicability
>
> We note that $\log^* (.)$ is an extremely slow-growing function; for example, $\log^*(x) \le 5$ for any $x \le 2^{65536}$. As a point of comparison, the number of atoms in the observable universe is estimated to be around $2^{265}$. Sample complexity is one of the main measures in context of replicability. Our work provides both upper and lower bounds on the sample complexity of the problem, thereby quantifying the price of replicability in this setting.
>
>
>
> If you feel that our response has adequately addressed your major concerns, we would appreciate it if you possibly adjust your score accordingly.
>
> **References**:
>
> [1] Pablo D Azar, Robert Kleinberg, and S Matthew Weinberg. Prophet inequalities with limited information. In SODA, 2014.
>
> [2] Aviad Rubinstein, Jack Z Wang, and S Matthew Weinberg. Optimal single-choice prophet inequalities from samples. Innovations in Theoretical Computer Science, 2020.
>
> [3] Jos´e Correa, Andr´es Cristi, Boris Epstein, and Jos´e A Soto. Sample-driven optimal stopping: From the secretary problem to the iid prophet inequality. Mathematics of Operations Research, 2023.
>
> [4] Jos´e Correa, Andr´es Cristi, Boris Epstein, and Jos´e Soto. The two-sided game of googol. The Journal of Machine Learning Research, 23(1):4870–4906, 2022.
>
> [5] Constantine Caramanis, Paul D¨utting, Matthew Faw, Federico Fusco, Philip Lazos, Stefano Leonardi, Orestis Papadigenopoulos, Emmanouil Pountourakis, and Rebecca Reiﬀenh¨auser. Single-sample prophet inequalities via greedy-ordered selection. In SODA, 2022.
>
> [6] Haim Kaplan, David Naori, and Danny Raz. Online weighted matching with a sample. In SODA, 2022.
>
> [7] Andrés Cristi, and Bruno Ziliotto. Prophet inequalities require only a constant number of samples. In STOC, 2024
>
> [8] Russell Impagliazzo, Rex Lei, Toniann Pitassi, and Jessica Sorrell. Reproducibility in learning. In STOC, 2022
>
> [9] Mark Bun, Marco Gaboardi, Max Hopkins, Russell Impagliazzo, Rex Lei, Toniann Pitassi, Satchit Sivakumar, and Jessica Sorrell. Stability is stable: Connections between
> replicability, privacy, and adaptive generalization. In STOC, 2023
>
> [10] Hossein Esfandiari, Amin Karbasi, Vahab Mirrokni, Grigoris Velegkas, and Felix Zhou. Replicable clustering. In NeurIPS, 2023.
>
> [11] Max Hopkins, Russell Impagliazzo, Daniel Kane, Sihan Liu, and Christopher Ye. Replicability in High Dimensional Statistics. In FOCS, 2024.
>
>
> [12] Mohammad Taghi Hajiaghayi, Robert Kleinberg, and Tuomas Sandholm. Automated online mechanism design and prophet inequalities. In AAAI, 2007.
>
> [13] Robert Kleinberg, and Matthew Weinberg. Matroid prophet inequalities. In STOC, 2012.
>
> [14] Michal Feldman, Nick Gravin, and Brendan Lucier. Combinatorial auctions via posted prices. In SODA, 2015.
>
> [15] Shuchi Chawla, Jason D. Hartline, David L. Malec, and Balasubramanian Sivan. Multi-parameter mechanism design and sequential posted pricing. In STOC, 2010.

---

> > ### Author Response · Authors · 2025-08-03
> >
> > Dear Reviewer,
> >
> > We would like to kindly check whether you have any remaining questions or concerns. We would be happy to provide further clarification. If not, we would be grateful if you could consider updating your score in light of the rebuttal.

---

> > ### Comment · Reviewer_5wxJ · 2025-08-03
> >
> > Thank you for your response to my comments. My original score reflects my evaluation of the paper, and I will keep it.

---

### Official Review · Reviewer_akGx · 2025-07-03

**Clarity:** 3
**Significance:** 3
**Originality:** 3
**Rating:** 5
**Confidence:** 3

**Summary:**

This paper explores the concept of replicability in online pricing problems, especially in the prophet inequalities and delegation problems settings. Replicability involves the stability of an algorithm with respect to different input data, which stands central to transparency in economic decision-making. The authors present a replicable and almost optimal approach to prophet inequalities, setting the sample complexity to $poly(log^\star(X))$, where X is the ground set of distributions. They use these findings on the delegation problem with an optimal matching lower bound setting the $\log^*|X|$ requirement. One notable technical improvement is a new algorithm for a heavy hitter variant with a nearly linear dependence on the inverse of the heavy hitter parameter that improves upon prior work by quite a margin.

**Questions:**

Could the authors suggest directions for future work in empirically analyzing the implemented replicable pricing strategies, through simulations on synthetic or real datasets to test how they perform and robustify under real-world conditions?

How could the results be extended to cover distributions whose support is continuous or of unknown support addressing real environment settings?

**Ethical Concerns:**

["NO or VERY MINOR ethics concerns only"]

**Final Justification:**

I haven't changed my opinion on this paper, I would be happy with acceptance.

**Limitations:**

Yes

**Quality:**

3

**Strengths And Weaknesses:**

Strengths:

The pricing strategy  proposed for prophet inequalities is shown to be attainable and nearly optimal, with a sample complexity polynomial in $\log^*|X|$, a significant improvement over the additive error solutions. In fact, matching lower bounds are provided.

The paper addresses the important problem of replicability in algorithms which is indeed of interest recently.

The paper introduces a novel algorithm for a variant of the heavy hitter problem with reduced sample complexity from cubic to effectively linear dependence on the inverse of the heavy hitter parameter. This is an important technical contribution in its own right.
The generalization of the replicable online pricing model and results to the delegation problem demonstrates the generality and broader applicability of the authors’ techniques.


Weaknesses

The main theorems (Theorem 1 and Theorem 2) assume the distributions have support on a given finite set $X$. While this is a typical starting assumption for theoretical computer science, it might limit straightforward application to continuous or unseen support spaces in practical use.

---

> ### Author Rebuttal · Authors · 2025-07-30
>
> Thank you for your positive review!
> Below we address your individual comments and questions.
>
> > .. suggest directions for future work in empirically analyzing the implemented replicable pricing strategies …
>
> We thank you for this suggestion and will include it in our paper as a direction for future work.
>
> > How could the results be extended to cover distributions whose support is continuous or of unknown support addressing real environment settings?
>
> We note that our lower bound in Theorem 3 shows that the problem cannot be solved with finite sample complexity for infinite sets. This is because a solution for an infinite set would yield a solution for arbitrarily large finite subsets, which by this theorem requires arbitrarily large sample complexity (see also footnote 3 in page 6)

---

> > ### Comment · Reviewer_akGx · 2025-08-09
> >
> > My opinion on this paper remains the same, thanks for the rebuttal.

---

### Decision · Program_Chairs · 2025-09-17

**Decision:**

Accept (poster)

**Comment:**

This paper explores the concept of replicability in online pricing problems, especially in the prophet inequalities and delegation problems settings. The paper has a set of concrete and solid results, and all reviewers appreciated the importance of the problem, the significance of the results, and the technical novelty of the theoretical results.

That said, there are still some concerns regarding the statements and presentation of the results, and more importantly, on what we know from previous works and what the contributions are in terms of techniques of this one. For instance, some reviewers were not satisfied with the answer because it didn't mention/cite the paper that establishes the median algorithm for the prophet inequality problem, as it is folklore in the community. Please consider addressing the presentation issues and proper citing of prior work in the final camera-ready version of your work.